# PARP1-dependent recruitment of the FBXL10-RNF68-RNF2 ubiquitin ligase to sites of DNA damage controls H2A.Z loading

**Gergely Rona[1,2†], Domenico Roberti[1,2†‡], Yandong Yin[1,2], Julia K Pagan[1,2§], Harrison Homer[1,2], Elizabeth Sassani[1,2], Andras Zeke[3], Luca Busino[1,2#¶], Eli Rothenberg[1,2], Michele Pagano[1,2,4]\***

[1]Department of Biochemistry and Molecular Pharmacology, New York University School of Medicine, New York, United States; [2]Perlmutter Cancer Center, New York University School of Medicine, New York, United States; [3]Institute of Enzymology, Research Center for Natural Sciences, Hungarian Academy of Sciences, Budapest, Hungary; [4]Howard Hughes Medical Institute, New York University School of Medicine, New York, United States

**\*For correspondence:**
michele.pagano@nyumc.org

[†]These authors contributed equally to this work

**Present address:** [‡]Department of Pediatrics, School of Medicine, University of Campania 'Luigi Vanvitelli', Naples, Italy; [§]School of Biomedical Sciences, Faculty of Medicine, The University of Queensland, Brisbane, Australia; [#]Department of Cancer Biology, University of Pennsylvania Perelman School of Medicine, Philadelphia, United States; [¶]Abramson Family Cancer Research Institute, University of Pennsylvania Perelman School of Medicine, Philadelphia, United States

**Competing interests:** The authors declare that no competing interests exist.

**Abstract** The mammalian FBXL10-RNF68-RNF2 ubiquitin ligase complex (FRRUC) mono-ubiquitylates H2A at Lys119 to repress transcription in unstressed cells. We found that the FRRUC is rapidly and transiently recruited to sites of DNA damage in a PARP1- and TIMELESS-dependent manner to promote mono-ubiquitylation of H2A at Lys119, a local decrease of H2A levels, and an increase of H2A.Z incorporation. Both the FRRUC and H2A.Z promote transcriptional repression, double strand break signaling, and homologous recombination repair (HRR). All these events require both the presence and activity of the FRRUC. Moreover, the FRRUC and its activity are required for the proper recruitment of BMI1-RNF2 and MEL18-RNF2, two other ubiquitin ligases that mono-ubiquitylate Lys119 in H2A upon genotoxic stress. Notably, whereas H2A.Z is not required for H2A mono-ubiquitylation, impairment of the latter results in the inhibition of H2A.Z incorporation. We propose that the recruitment of the FRRUC represents an early and critical regulatory step in HRR.
DOI: https://doi.org/10.7554/eLife.38771.001

## Introduction

Polycomb group (PcG) proteins assemble into heterogeneous Polycomb Repressive Complexes (PRCs), of which there are two main, functionally distinct classes, each with different histone modifying activities. PRC1s mediate the mono-ubiquitylation of histone H2A at Lys119 (K119) (generating H2AK119Ub1), whereas PRC2s di- and tri-methylate K27 of histone H3 (H3K27me2/3) (*Blackledge et al., 2015*; *Di Croce and Helin, 2013*; *Schwartz and Pirrotta, 2013*).

There are six PRC1s, each defined by one of six different Polycomb group RING finger proteins (PCGF1-6), which belong to the super-family of RING finger (RNF) ubiquitin ligases (*Gao et al., 2012*). Although PCGFs have a RING finger domain, they are unable to ubiquitylate H2A; instead, PCGFs dimerize with and activate one of two additional RNF ubiquitin ligases, RNF1 or RNF2, (also known as RING1A and RING1B, respectively), which in turn ubiquitylate H2A (*Cao et al., 2005*; *de Napoles et al., 2004*; *Wang et al., 2004*). Based on the presence or absence of chromobox-containing protein (CBX) subunits, the six PRC1s are further sub-categorized into two canonical (cPRC1) and four non-canonical (ncPRC1) complexes, respectively. The best-characterized ncPRC1 is

ncPRC1.1 (also known as variant PRC1.1), which contains a core ubiquitin ligase complex, the FRRUC, which comprises the PCGF protein RNF68 (also known as PCGF1 and NSPC1), RNF2, and the short isoform (which misses the JmjC domain) of the F-box protein FBXL10 (also known as FBL10, JHDM1B, KDM2B, and NDY1) (*Gearhart et al., 2006*; *Inagaki et al., 2015*; *Oliviero et al., 2015*; *Sánchez et al., 2007*; *Wu et al., 2013*).

The FRRUC mono-ubiquitylates H2A on K119, repressing genes that are important for differentiation and development (*Cao et al., 2005*; *Endoh et al., 2012*; *Gearhart et al., 2006*; *He et al., 2008*; *Wang et al., 2004*; *Wu et al., 2013*). Mono-ubiquitylation of K119 in H2A and K120 in H2A.X has also been detected at DNA lesions such as double-strand breaks (DSBs) (*Bergink et al., 2006*; *Ginjala et al., 2011*; *Kakarougkas et al., 2014*; *Pan et al., 2011*; *Shanbhag et al., 2010*; *Ui et al., 2015*). The DSB-induced mono-ubiquitylation of H2A is thought to promote transcriptional repression at sites of DSBs (*Kakarougkas et al., 2014*; *Shanbhag et al., 2010*; *Ui et al., 2015*). Two dimeric ubiquitin ligases (BMI1-RNF2 and MEL18-RNF2 also know as RNF51-RNF2 and RNF110-RNF2, respectively), which are components of two distinct cPRC1s, have been implicated in the mono-ubiquitylation of K119 and K120 in H2A and H2A.X in response to genotoxic stress (*Bergink et al., 2006*; *Ginjala et al., 2011*; *Gracheva et al., 2016*; *Ismail et al., 2010*; *Pan et al., 2011*). However, whether ncPRC1s, which in transcription regulation are required for the ubiquitylation of H2A via cPRC1s (*Blackledge et al., 2014*, *2015*; *Cooper et al., 2014*), are also necessary in response to DNA damage has remained unknown.

We found that the FRRUC is rapidly and transiently recruited to sites of DNA damage in a PARP1- and TIMELESS-dependent manner and determined the biological significance of this recruitment as described below.

## Results

### The FRRUC is rapidly recruited to sites of DNA damage in a PARP1- and TIMELESS-dependent manner, promoting the subsequent recruitment of cPRC1s

FBXL10, RNF68, and RNF2 are three subunits of a ubiquitin ligase complex, named FRRUC for short (*Gearhart et al., 2006*; *Sánchez et al., 2007*; *Wu et al., 2013*). We confirmed that the three proteins exist in a trimeric complex by performing sequential co-precipitation experiments in which we found that FBXL10-interacting RNF2 also interacted with RNF68 (*Figure 1—figure supplement 1A*). In unstressed cells, mono-ubiquitylation of K119 in H2A via the FRRUC, which serves as the core ubiquitin ligase of ncPRC1.1, is required for the recruitment of cPRC1s to maintain the mono-ubiquitylation of H2A (*Blackledge et al., 2015*; *Cooper et al., 2014*). Certain cPRC1s have been shown to be recruited to DNA lesions and mono-ubiquitylate H2A and H2A.X (*Bergink et al., 2006*; *Ginjala et al., 2011*; *Gracheva et al., 2016*; *Ismail et al., 2010*; *Pan et al., 2011*). Therefore, we asked whether the FRRUC is also recruited. Local DNA damage was introduced into human U-2OS cells by 405 nm laser micro-irradiation, as previously described (*Altmeyer et al., 2015*; *Young et al., 2015*). The track of DNA damage was then visualized by γH2A.X staining. We found that endogenous FBXL10, RNF68, and RNF2 recruited to sites of DNA damage and co-localized with γH2A.X (*Figure 1—figure supplement 1B*).

To use a different imaging method that is especially advantageous for detecting and quantifying localization patterns and protein complexes in dense structures, such as nuclei (*Ricci et al., 2015*; *Yin and Rothenberg, 2016*), we used dual-color single-molecule localization microscopy (direct STochastic Optical Reconstruction Microscopy or dSTORM) to evaluate the enrichment of the FRRUC at DNA damage sites in U-2OS cells upon treatment with the radiomimetic agent, neocarzinostatin (NCS), which induces single and double stranded breaks. As a DNA marker we used XRCC5, a highly abundant protein that is rapidly recruited to DSBs since, although it has intrinsic affinity for DNA, it has even higher affinity for DNA ends. [XRCC5 is recruited very rapidly before the choice of homologous recombination repair (HHR) v. non-homologous end joining (NHEJ) is made by the cell.] Because it associates with chromatin in both undamaged and damaged cells, XRCC5 is an ideal marker to evaluate cross-correlation changes with other DNA-binding proteins in response to DNA damage (*Britton et al., 2013*; *Reid et al., 2015*; *Teixeira-Silva et al., 2017*; *Walker et al., 2001*). *Figure 1A* shows representative overlaps between FBXL10, RNF68, or RNF2 and XRCC5 with or

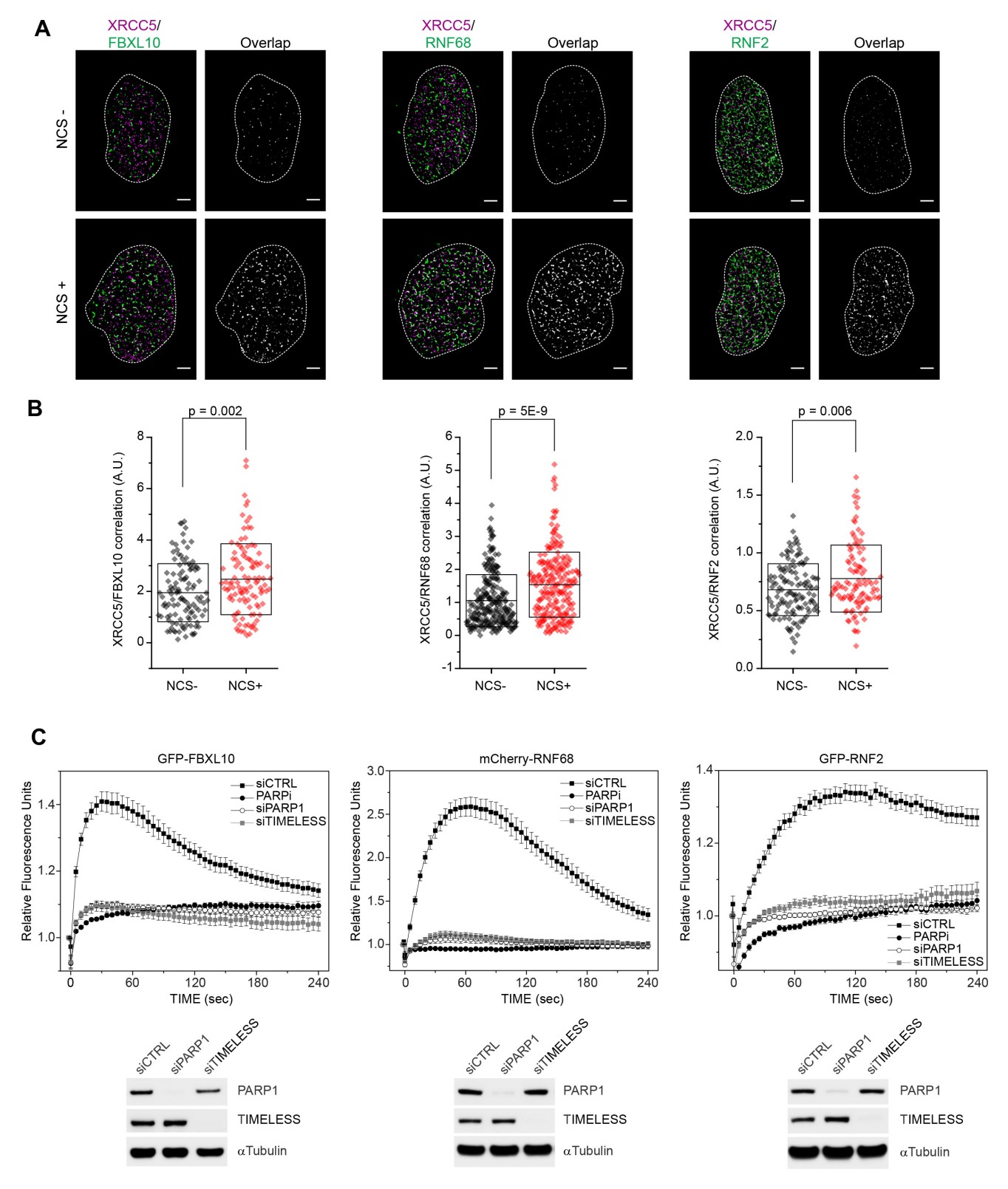

**Figure 1.** FBXL10, RNF68, and RNF2 are recruited to sites of DNA damage in a PARP1- and TIMELESS- dependent manner. (**A**) Super-resolution imaging was performed in U-2OS cells to analyze the colocalization between XRCC5 (magenta) and FBXL10, RNF68 and RNF2 (green), with or without 20 min of NCS treatment. Representative nuclei are shown. A white dash line denotes the border of each nucleus. Scale bar represents 2.5 μm. The signal overlap (white) demonstrates the colocalization between XRCC5 and FBXL10, RNF68 or RNF2. (**B**) The graphs display the correlation between

*Figure 1 continued on next page*

*Figure 1 continued*

XRCC5 and FBXL10 (NCS-, n = 113; NCS+, n = 109), RNF68 (NCS-, n = 254; NCS+, n = 232) or RNF2 (NCS-, n = 128; NCS+, n = 105) in individual nuclei, respectively. Each plot is the correlation amplitude between XRCC5 and FBXL10/RNF68/RNF2 within one nucleus. The n value representing the total number of analyzed nuclei is pooled from the three or four independent experiments. A.U., arbitrary units. Whisker box shows mean and S.D. values. *P* values were calculated using two-sample t-test between NCS - and NCS + samples. (C) U-2OS cells stably expressing GFP-FBXL10 (left), mCherry-RNF68 (middle) or GFP-RNF2 (right) were transfected with siRNAs targeting PARP1, TIMELESS, or a non-targeting control (CTRL). Cells were pre-sensitized with BrdU (10 μM) for 36 hr and subjected to 405 nm laser induced damage. Where indicated, cells were pretreated with 1 μM PARP inhibitor (Olaparib) for 1 hr. DNA damage recruitment dynamics were captured by live cell imaging. Relative fluorescence values and images were acquired every 5 s for 4 min. For each condition, ≥25 cells were evaluated from 2 or three independent experiments. Mean relative fluorescence values and standard errors were plotted against time. Representative images are shown in *Figure 1—figure supplement 2A*. Times are indicated in seconds. The efficiency of PARP1 and TIMELESS depletion is shown using immunoblotting.

DOI: https://doi.org/10.7554/eLife.38771.002

The following figure supplements are available for figure 1:

**Figure supplement 1.** The trimeric FRUCC recruits to sites of DNA damage.

DOI: https://doi.org/10.7554/eLife.38771.003

**Figure supplement 2.** Recruitment of the FRUCC to DNA damage sites.

DOI: https://doi.org/10.7554/eLife.38771.004

**Figure supplement 3.** Extended kinetics of FBXL10 recruitment, and TIMELESS-independent recruitment of XRCC1 and Ligase 3.

DOI: https://doi.org/10.7554/eLife.38771.005

without NCS treatment. Since in dense images, it is possible that two species randomly overlap with each other, we analyzed all images using a cross-correlation method that probes the probability distributions across all possible pair-wise distances between two species taking in account, at the same time, the amount of each species (*Coltharp et al., 2014*; *Veatch et al., 2012*). We observed a significant increase in cross-correlation signal between FBXL10, RNF68, or RNF2 and XRCC5, upon NCS treatment (*Figure 1B*), showing an enrichment of the FRRUC at sites of DNA damage in agreement with the results obtained with laser micro-irradiation.

To examine the spatiotemporal dynamics of FBXL10, RNF68, and RNF2 after DNA damage in living cells, we generated cells stably expressing fluorescent protein-tagged versions of these three proteins under the control of a weak LTR promoter. In living cells, FBXL10 was localized to the nucleoplasm with an enrichment in the nucleolus (*Figure 1—figure supplement 2A*), as previously reported both in mammalian and insect cells (*Frescas et al., 2007*; *Kavi and Birchler, 2009*). Whereas RNF68 displayed a localization similar to that of FBXL10, RNF2 showed a more homogenous nuclear distribution (*Figure 1—figure supplement 2A*), likely due to the fact that RNF2 can participate in all six PRC1s. Often DDR proteins are sequestered into the nucleolus and redistributed to sites of DNA repair in response to genotoxic stress (*Antoniali et al., 2014*; *Lindström et al., 2018*). When we tested the recruitment to laser-induced 'DNA damage spots' (*Young et al., 2015*), we found that GFP-FBXL10, mCherry-RNF68, and GFP-tagged RNF2 were recruited within seconds to these spots (*Figure 1C*). The peak of recruitment was reached by approximately within one minute post-irradiation, after which the signal progressively decreased (*Figure 1C*). *Figure 1—figure supplement 2A* shows representative examples of recruitment of these three proteins. We noticed that whereas the kinetics of disappearance of FBXL10 and RNF68 were comparable, RNF2 remained at DNA lesions for longer, likely because of its presence in both cPRC1s and ncPRC1s. Notably, no recruitment of FBXL11 (also known as FBL11, JHDM1A, and KDM2A), a close paralog of FBXL10 that does not bind RNF2 and RNF68 (*Figure 1—figure supplement 2B*), was observed (*Figure 1—figure supplement 2C*), suggesting that FBXL10 plays a specific role in the DDR.

To test whether the recruitment of the FRRUC to sites of DNA damage depends on known DNA damage signaling enzymes, we evaluated the recruitment of the three subunits upon treatment of cells with small molecules to inhibit ATM (with KU60018), ATR (with AZ20), DNA-PK (with NU7441), and PARP (with Olaparib)(*Figure 1C* and *Figure 1—figure supplement 2A,D*). Only the inhibition of PARP [Poly(ADP-ribose) polymerase] with Olaparib completely abolished the recruitment of the FRRUC to sites of DNA damage (*Figure 1C* and *Figure 1—figure supplement 2A,D*). We did not observe recruitment of FBXL10 upon PARP inhibition up to 30 min after laser micro-irradiation (*Figure 1—figure supplement 3A*). Importantly, inhibition of PARP with Olaparib also abolished the recruitment of endogenous proteins (*Figure 1—figure supplement 1B*). PARP1 is a critical sensor of

DNA damage lesions, catalyzing the production of long chains of branched poly(ADP)ribose on target proteins during the DNA damage response (DDR) (*Li and Yu, 2015*). PARylation is one of the earliest post-translational modifications in response to DNA lesions, occurring on substrates such as histones, certain DNA repair factors, and PARP1 itself (*Rouleau et al., 2010*). We, and others, have shown that TIMELESS forms a near stoichiometric complex with PARP1 (*Xie et al., 2015*; *Young et al., 2015*). Therefore, we monitored the recruitment of the FRRUC to sites of DNA damage after silencing either PARP1 or TIMELESS, finding that the recruitment of all three proteins was abolished after depletion of either PARP1 or TIMELESS (*Figure 1C* and *Figure 1—figure supplement 2A*). In contrast, the recruitment of other DNA repair factors, such as XRCC1 and Ligase 3, which are affected by PARP1 silencing (*Campalans et al., 2013*; *Godon et al., 2008*; *Mortusewicz et al., 2006*; *Okano et al., 2003*), were not affected by TIMELESS depletion (*Figure 1—figure supplement 3B*).

Collectively, our results demonstrate that the PARP1-TIMELESS complex promotes the rapid recruitment of the FRRUC to sites of DNA damage, suggesting an early role for this ubiquitin ligase complex in the response to DNA damage.

Since two dimeric ubiquitin ligases (RNF51-RNF2 and RNF110-RNF2), which are component of two distinct cPRC1s, have been shown to recruit to DNA lesions and mono-ubiquitylate H2A (*Bergink et al., 2006*; *Ginjala et al., 2011*; *Gracheva et al., 2016*; *Ismail et al., 2010*; *Pan et al., 2011*), we asked whether the recruitment of the FRRUC depends on the recruitment of these cPRC1 components. We found that the recruitment of FBXL10 and RNF68 was not affected by the depletion of either RNF51 (also called BMI1 or PCGF4) or RNF110 (also called MEL18 or PCGF2) (*Figure 2A*-and *Figure 2—figure supplement 1A*). The recruitment of RNF2 was partially inhibited by the depletion of either RNF51, RNF110, or RNF68, likely because of its presence in both cPRC1s and ncPRC1s. Both GFP-tagged RNF51 and RNF110 recruited to laser induced DNA damage sites which peeks approximately around ~75 s post-irradiation and they remain at DNA lesions longer than FBXL10 and RNF68. We found that the recruitment of RNF51 and RNF110 depends on PARP1 and TIMELESS presence as well as PARP1 activity (*Figure 2B–C*-and *Figure 2—figure supplement 1B*).

To test whether the FRRUC regulates the recruitment of RNF51 and RNF110, we silenced RNF68 since RNF68 is the specific PCGF subunit that defines ncPRC1.1 and has not been identified in other protein complexes as FBXL10 and RNF2 (*Gao et al., 2012*; *Sánchez et al., 2007*; *Wong et al., 2016*). We found that the recruitment of RNF51 and RNF110 was inhibited by the depletion of RNF68 (*Figure 2B–C*- and *Figure 2—figure supplement 1B*). To control for off-target effects, we set to rescue the recruitment of RNF51 and RNF110 by complementation with siRNA-resistant constructs of RNF68. In addition to wild-type RNF68, we sought to use inactive RNF68 mutants; so, we used a panel of truncation mutants to identify the minimal region that supports the interaction of RNF68 with FBXL10 and RNF2. We found that four amino acids at position 251–254 of human RNF68 (QYSV) are essential for a stable binding with FBXL10 (*Figure 2—figure supplement 2*). Further refinement of the interaction surface between RNF68 and FBXL10 indicated that mutation of tyrosine 252 to aspartic acid [generating RNF68(Y252D)] significantly reduced RNF68's binding to FBXL10 without affecting its interaction with RNF2 (*Figure 2—figure supplement 2*). In contrast, mutation of Y252 to alanine [generating RNF68(Y252A)] had no effect on RNF68's interaction with FBXL10. Deletion of the RING domain of RNF68 [creating RNF68(ΔRING)] completely abrogated its binding to RNF2, while leaving its interaction with FBXL10 intact (*Figure 2—figure supplement 2*), indicating that RNF2 is not required for the interaction between FBXL10 and RNF68, in agreement with the crystal structure of the minimal version of ncPRC1.1 (*Wong et al., 2016*).

We found that wild-type RNF68, but not RNF68 inactive mutants, partially prevented the inhibition of RNF51's and RNF110's recruitment due to RNF68 silencing (*Figure 2B–C* and *Figure 2—figure supplement 1B*). These results demonstrate that RNF68's interaction with both RNF2 and FBXL10, which is required form an active FRRUC, is necessary for the proper recruitment of RNF51 and RNF110.

Thus, it appears that an active FRRUC is necessary for the proper recruitment of cPRC1 complexes, similarly to what was previously shown for transcriptional repression in unperturbed cells.

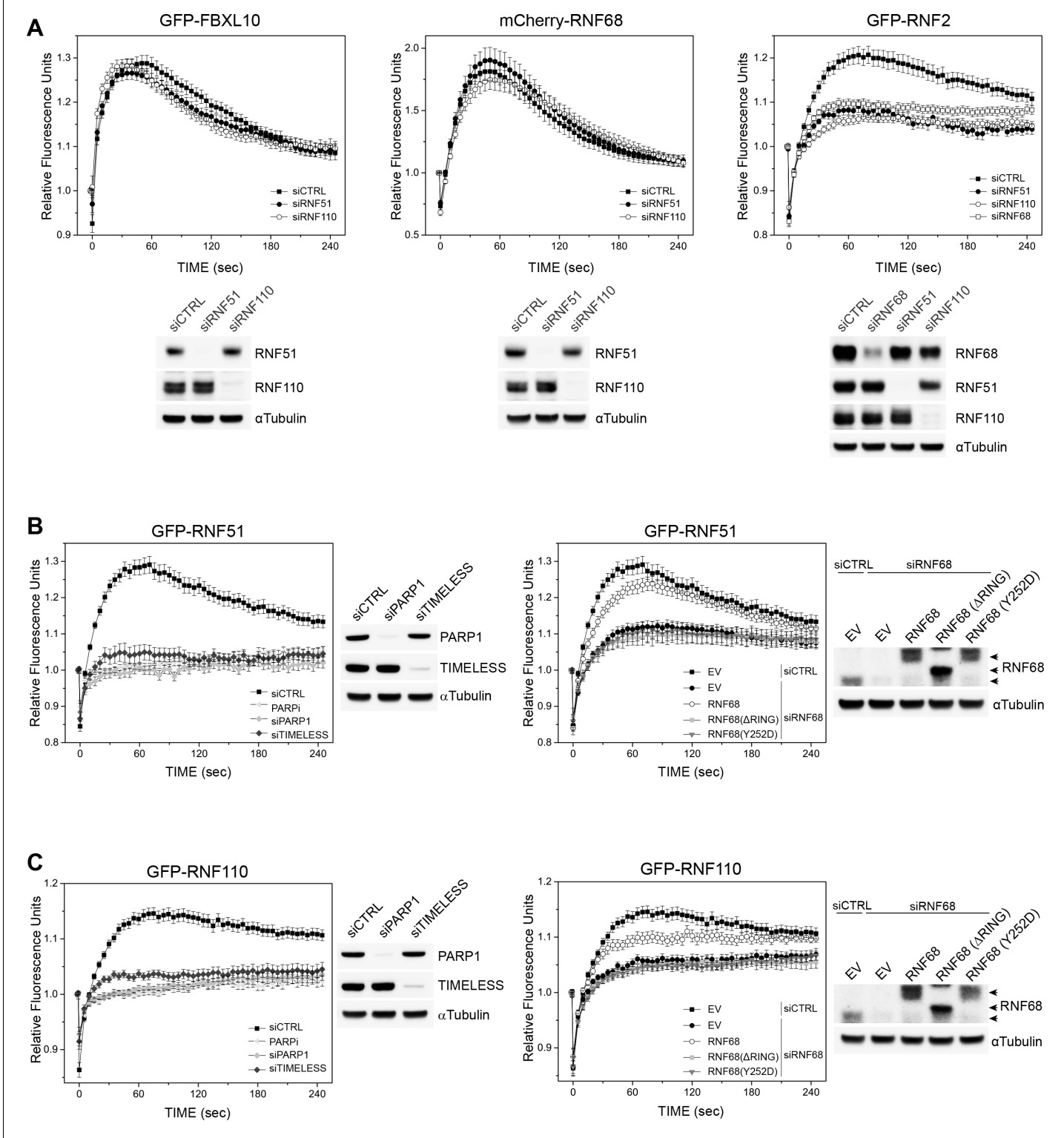

**Figure 2.** PARP1-, TIMELESS-, and RNF68-dependent recruitment of the cPRC1 subunits, RNF51 and RNF110. (**A**) U-2OS cells stably expressing GFP-FBXL10 (left), mCherry-RNF68 (middle) or GFP-RNF2 (right) were transfected with siRNAs targeting RNF51, RNF110 or a non-targeting control (CTRL). Cells were pre-sensitized with BrdU (10 μM) for 36 hr and subjected to 405 nm laser induced damage. DNA damage recruitment dynamics were captured by live cell imaging. Relative fluorescence values and images were acquired every 5 s for 4 min. For each condition,≥35 cells were evaluated from 2 or three independent experiments. Mean relative fluorescence values and standard errors were plotted against time. Representative images are shown in **Figure 2—figure supplement 1A**. Times are indicated in seconds. The efficiency of RNF51, RNF110 and RNF68 depletion is shown using

*Figure 2 continued on next page*

*Figure 2 continued*

immunoblotting. (**B**) Left: U-2OS cells stably expressing GFP-RNF51 were transfected with siRNAs targeting PARP1, TIMELESS or a non-targeting control (CTRL). Right: U-2OS cells engineered to express FLAG-HA-tagged RNF68, RNF68(ΔRING) or RNF68(Y252D) in combination with GFP-RNF51 were transfected with siRNAs targeting the 3' UTR of RNF68 (siRNF68 seq2). Cells were pre-sensitized with BrdU (10 µM) for 36 hr and subjected to 405 nm laser induced damage. Where indicated, cells were pretreated with 1 µM PARP inhibitor (Olaparib) for 1 hr. DNA damage recruitment dynamics were captured by live cell imaging. Relative fluorescence values and images were acquired every 5 s for 4 min. For each condition, ≥35 cells were evaluated from 2 or three independent experiments. Mean relative fluorescence values and standard errors were plotted against time. CTRL sample set is the same on both graphs. Representative images are shown in *Figure 2—figure supplement 1B*. Times are indicated in seconds. The efficiency of PARP1, TIMELESS and RNF68 depletion and over-expression of the FLAG-HA-tagged RNF68 rescue constructs is shown using immunoblotting. Upper arrow indicates FLAG-HA-tagged RNF68 and RNF68(Y252D), middle arrow indicates RNF68(ΔRING) while lower arrow corresponds to endogenous RNF68. (**C**) Same as in (**B**) but using U-2OS cells stably expressing GFP-RNF110.

DOI: https://doi.org/10.7554/eLife.38771.006

The following figure supplements are available for figure 2:

**Figure supplement 1.** Representative images for *Figure 2*.
DOI: https://doi.org/10.7554/eLife.38771.007

**Figure supplement 2.** Mapping of FBXL10 and RNF2 binding domains in RNF68.
DOI: https://doi.org/10.7554/eLife.38771.008

## The FRRUC mediates mono-ubiquitylation of H2A on K119 at sites of DNA damage

Since the FRRUC mono-ubiquitylates H2A on K119 in unstressed cells, we monitored the ability of FBXL10, RNF68, and RNF2 to regulate the ubiquitylation of H2A on Lys119 at sites of laser-induced DNA damage. Knocking down either FBXL10, RNF68, or RNF2 (*Figure 3—figure supplement 1A*) resulted in a significant reduction of H2AK119Ub1 staining on the laser damage tracks compared to control samples (*Figure 3A*). Likewise, silencing PARP1 or TIMELESS, which are required for proper localization of the complex, or inhibition of PARP, resulted in a decrease in H2AK119Ub1 levels at sites of DNA damage (*Figure 3A*).

In order to control for off-target effects, we sought to rescue the H2AK119Ub1 levels phenotype by complementation of siRNA-resistant constructs. We focused on rescuing the effect of RNF68 depletion since RNF68 is the specific PCGF subunit that defines the ncPRC1.1 and because RNF68 is the only component of the FRRUC that does not appear to be present in other protein complexes (*Gao et al., 2012*; *Sánchez et al., 2007*; *Wong et al., 2016*). We found that, in contrast to the siRNA-resistant wild type RNF68, neither the siRNA-resistant RNF68(ΔRING) nor the siRNA-resistant RNF68(Y252D) could rescue the defects in H2AK119Ub1 levels at DNA damage sites (*Figure 3B*).

Next, we examined the relative levels of H2AK119Ub1 over H2A at defined DSBs. To this end we utilized a DSB reporter system in U-2OS cells in which DSBs could be generated at a defined genomic locus based on the inducible expression of an ER-mCherry-LacI-FokI-DD nuclease fusion protein (*Shanbhag et al., 2010*; *Tang et al., 2013*). This cell line contains the FokI nuclease fused to both a modified estradiol receptor (ER) and a destabilization domain (DD), allowing its inducible nuclear translocation and stabilization after treatment with 4-hydroxytamoxifen (4-OHT) and Shield-1, respectively (*Figure 3C*). The nuclease then localizes to a lac operon repeat stably integrated into the genome of these engineered U-2OS cells to induce site specific DSBs within the region. Using qChIP with specific primers for a region downstream to the DSBs, we observed that levels of H2AK119Ub1 were induced ~4.2 kbp from the break sites, and to a slightly lesser extent ~10.2 kbp, from the DSB sites (*Figure 3C*). Depletion of FBXL10, RNF2, or RNF68 (but not of XRCC6, used as a negative control) strongly reduced the amount of H2AK119Ub1 detected at the sites of DSBs (*Figure 3C* and *Figure 3—figure supplement 1B*), indicating that these proteins significantly contribute to the mono-ubiquitylation of H2A at K119.

## The FRRUC promotes transcriptional repression in response to DNA damage

The H2AK119Ub1 modification has been strongly associated with transcriptional silencing at DSBs (*Kakarougkas et al., 2014*; *Sanchez et al., 2016*, *2010*; *Ui et al., 2015*). Having confirmed that the FRRUC controls the mono-ubiquitylation of H2A on K119 at sites of DNA damage, we examined whether the disruption of the FRRUC affects DSB-mediated transcriptional repression. To this end,

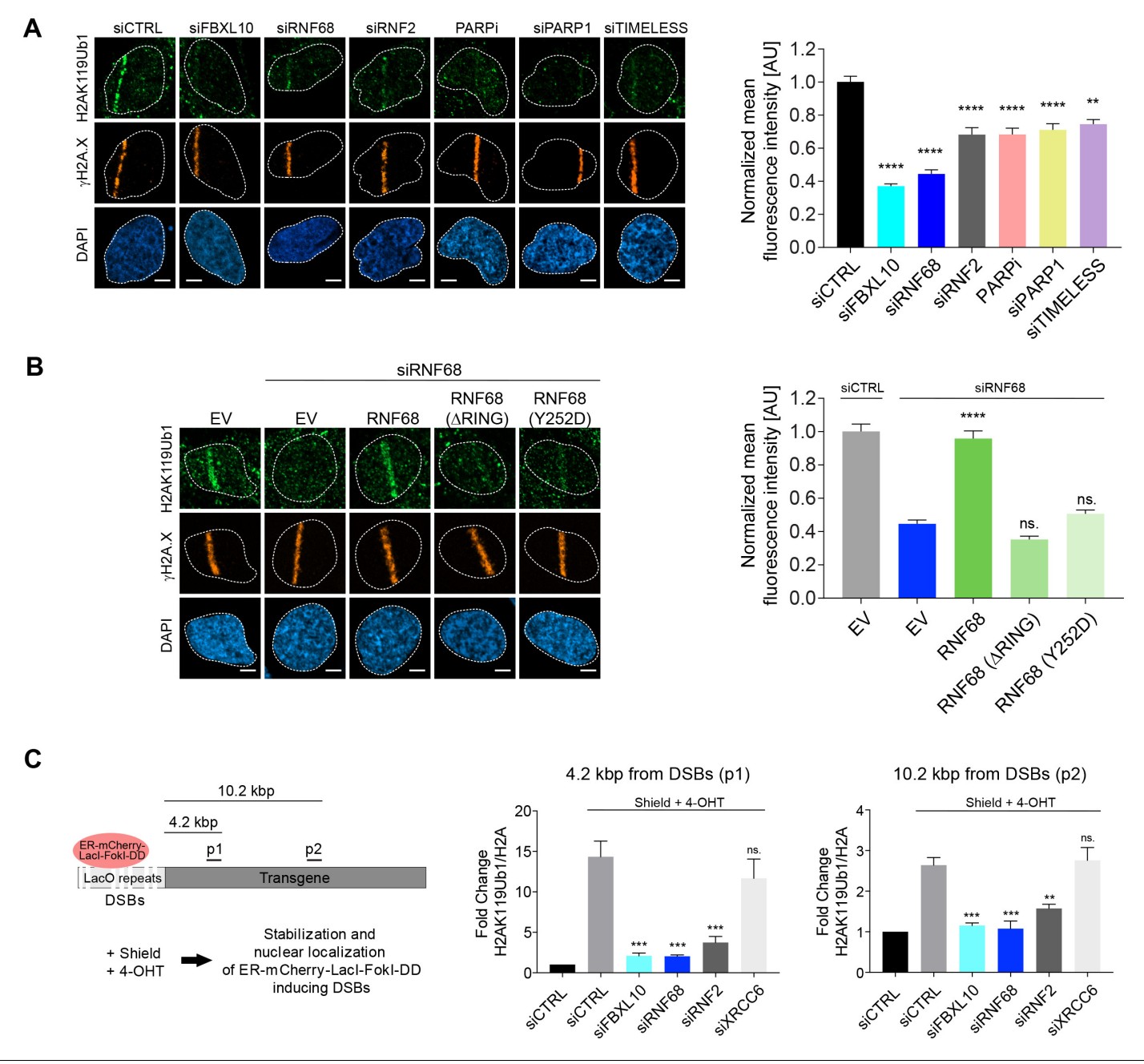

**Figure 3.** The FRRUC is required for H2A-ubiquitylation at sites of DNA damage. (**A**) U-2OS cells were transfected with the indicated siRNAs. Cells were fixed 30 min after laser microirradiation and stained with antibodies to γH2A.X (orange) and H2AK119Ub1 (green). Scale bar represents 5 μm. The graph represents the mean intensity of the H2AK119Ub1 signal on the γH2A.X track, normalized to the mean values of the siCTRL sample. At least 28 laser stripes were analyzed per condition from 2 or three independent experiments. A white dash line denotes the border of each nucleus. Error bars represent S.E.M. Differences relative to siCTRL were calculated using one-way ANOVA. ****p≤0.0001, ***p≤0.001, **p≤0.01, *p≤0.05, ns = not significant. (**B**) U-2OS cells engineered to express FLAG-HA-tagged RNF68, RNF68(ΔRING) or RNF68(Y252D) were transfected with siRNAs targeting the 3' UTR of RNF68 (siRNF68 seq2). Cells were fixed 30 min after laser microirradiation and stained with antibodies to γH2A.X (orange) and H2AK119Ub1 (green). Scale bar represents 5 μm. The graph represents the mean intensity of the H2AK119Ub1 signal on the γH2A.X track, normalized to the mean values of the siCTRL sample. At least 30 laser stripes were analyzed per condition from 2 or three independent experiments. A white dash line denotes the border of each nucleus. Error bars represent S.E.M. Differences relative to siCTRL were calculated using one-way ANOVA. ****p≤0.0001, ***p≤0.001, **p≤0.01, *p≤0.05, ns = not significant. (**C**) Left: Schematic of DSB generation by ER-mCherry-LacI-FokI-DD at lac operator (LacO) repeats integrated into the genome of U-2OS DSB reporter cells. The ER-mCherry-LacI-FokI-DD is stabilized by Shield1 and targeted to the nucleus by 4-OHT. The qChIP primer sets around the induced DSBs are labeled as p1-p2. ER, estrogen receptor; DD, destabilization domain. The ER-mCherry-LacI-FokI-

*Figure 3 continued on next page*

*Figure 3 continued*

DD nuclease was induced for 1 hr using Shield-1 and 4-OHT. Right: qChIP was performed in FokI reporter U-2OS cells with and without FokI-induced DSBs using antibodies to H2A and ubiquitylated H2A. To determine the relative amount of mono-ubiquitylated H2A, H2AK119Ub1 values were normalized on total H2A values. Error bars represent S.E.M from ≥3 independent experiments. *P* values were calculated using one-way ANOVA. ****p≤0.0001, ***p≤0.001, **p≤0.01, *p≤0.05, ns = not significant.

DOI: https://doi.org/10.7554/eLife.38771.009

The following figure supplement is available for figure 3:

**Figure supplement 1.** Representative efficiency of siRNA silencing for all oligos used in this study.

DOI: https://doi.org/10.7554/eLife.38771.010

we used a previously described transcriptional reporter system (*Janicki et al., 2004*; *Shanbhag et al., 2010*). Similar to the experiment shown in *Figure 3C*, DSBs were induced by activating the ER-mCherry-LacI-FokI-DD fusion protein. In addition, as schematized in *Figure 4A*, doxycycline (DOX) treatment enabled the transcription of CFP-SKL (CFP-tagged peroxisomal targeting peptide) mRNA that contains MS2 stem loop repeats. The latter, when bound by the stably transfected YFP-tagged MS2 viral coat protein (YFP-MS2), is detected as a yellow dense dot in the nucleus whose fluorescence intensity is quantified by ImageJ. The production of nascent RNA transcript was almost completely abolished in control cells upon the activation of the FokI nuclease (*Figure 4B*), indicative of transcriptional repression at sites of DNA damage. As previously reported (*Shanbhag et al., 2010*), ATM inhibition significantly restored transcription near the DSB sites. Depletion or mislocalization of the FRRUC (the latter accomplished through inhibition of PARP or depletion of either PARP1 or TIMELESS) also significantly de-repressed transcription near sites of DSB (*Figure 4B* and *Figure 4—figure supplement 1A*). Silencing of FBXL10, RNF68, RNF2, PARP1, or TIMELESS did not have a negative effect on FokI nuclease levels (*Figure 4—figure supplement 1A*).

As a complementary technique, we monitored nascent RNA transcript production at laser microirradiation-induced DNA damage sites using 5-ethynyl uridine (5-EU) labeling (*Jao and Salic, 2008*), which was evaluated as shown in *Figure 4—figure supplement 1B* (see also schematics of the experiment in *Figure 4C*). Nascent RNA transcript production at DNA damage stripes was significantly reduced in control cells (*Figure 4D*). By contrast, nascent RNA production was significantly less reduced at sites of DNA damage in cells in which the FRRUC was either depleted by siRNA or mislocalized by inhibiting PARP1 or by depleting either PARP1 or TIMELESS (*Figure 4D*).

Loss of nascent RNA production after DNA damage could be due to the stalling of RNA polymerase II (RNAPII). Therefore, we monitored the presence of actively elongating RNAPII at the laserinduced damage sites using an antibody against the phosphorylated heptapeptide repeats (Ser2) of the C-terminal domain of the RNAPII's large subunit (*Hsin and Manley, 2012*), as shown in the schematic (*Figure 4C*). As expected, phosphorylation of RNAPII on Ser2 was impaired at break sites in control cells but was significantly less impaired in cells in which FBXL10, RNF68 or RNF2 were depleted, or in which the complex was mislocalized by either inhibiting PARP or depleting PARP1 or TIMELESS (*Figure 4E*).

Altogether, these results suggest that the FRRUC is required for transcriptional silencing after DNA damage.

## The FRRUC promotes DNA damage signaling and homologous recombination repair

Because of its recruitment to DNA lesions, we hypothesized that the FRRUC might regulate the response to DNA damage. Therefore, we individually downregulated FBXL10, RNF68, and RNF2 in U-2OS cells and assessed their role in the recruitment of several DNA repair proteins, using RNF8 downregulation as a positive control (see Discussion) and NCS or camptothecin (CPT, a DNA topoisomerase one inhibitor) as DNA damaging agents (*Figure 5A–B*). The percentages of cells containing DNA-damage induced foci were evaluated as previously described (*Bunting et al., 2010*; *Escribano-Díaz et al., 2013*; *Orthwein et al., 2015*). *Figure 5—figure supplement 1A–B* shows representative examples of foci positive and foci negative cells. The silencing of FBXL10, RNF68, or RNF2 resulted in a significant reduction in the number of cells having DNA damage-induced foci containing the RAD51 recombinase (*Figure 5A–B* and *Figure 5—figure supplement 1A–B*). At the

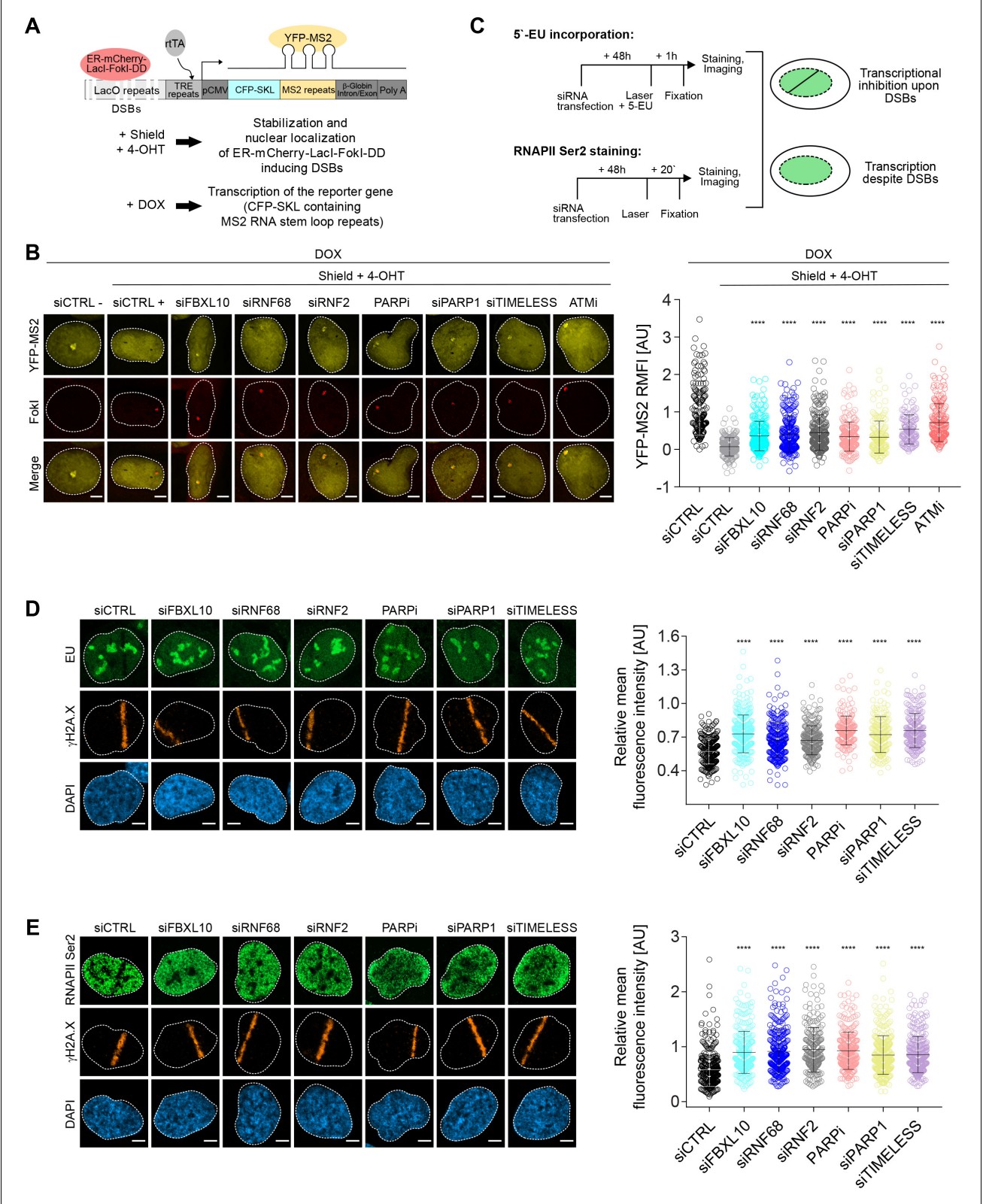

**Figure 4.** The FRUCC is required for transcriptional repression at sites of DNA damage. (**A**) Schematic of the U-2OS reporter system that allows monitoring transcription following induction of DSBs. DSBs are induced in the LacO repeats by activating the ER-mCherry-LacI-FokI-DD fusion protein with Shield-1 and 4-OHT. The transcription of the MS2 stem loop containing CFP-SKL is driven by a minimal CMV promoter and the binding of the tetracycline-controlled transactivator (rtTA) after DOX treatment. Nascent transcript is visualized by the binding of the YFP-MS2 viral coat protein. (**B**)
*Figure 4 continued on next page*

*Figure 4 continued*

Transcription activity was measured in U-2OS FokI-YFP-MS2 reporter cells transfected with the indicated siRNAs or inhibitors upon induction of DSBs. Quantification of YFP-MS2 relative mean fluorescence intensity (RMFI) was carried out from 3 to 4 independent experiments as in (*Shanbhag et al., 2010*). Error bars represent S.D. n = 179, 204, 280, 235, 263, 215, 148, 133 and 185 for siCTRL -, siCTRL +, siFBXL10, siRNF68, siRNF2, PARPi (Olaparib), siPARP1, siTIMELESS and ATMi (KU60018) respectively. Scale bar represents 5 µm. A white dash line denotes the border of each nucleus. *P* values were calculated using one-way ANOVA. ****p≤0.0001, ***p≤0.001, **p≤0.01, *p≤0.05, ns = not significant. (C) Schematics of 5-EU and RNAPII Ser2 staining to monitor transcription following laser micro-irradiation in U-2OS cells. Upon transcriptional silencing due to DNA damage, one can observe a dark anti-stripe for the 5-EU and RNAPII Ser2 staining (green) on the laser track which is highlighted by the γH2A.X staining. (D) U-2OS cells transfected with the indicated siRNAs were laser micro-irradiated and treated with 5-Ethynyl Uridine (5-EU) for 1 hr prior to fixation. Cells were stained with an antibody to γH2A.X (orange). 5-EU was detected using Click -IT imaging kit (green). Scale bar represents 5 µm. The dot plot represents the relative mean intensity of the 5-EU signal along the laser stripe compared to unaffected regions of the nucleus with error bars representing S.D. At least 20 cells were analyzed per condition from 2 or three independent experiments. A white dash line denotes the border of each nucleus. *P* values were calculated using one-way ANOVA. ****p≤0.0001, ***p≤0.001, **p≤0.01, *p≤0.05, ns = not significant. (E) U-2OS cells were fixed 20 min after laser micro-irridiation and stained with an antibody to γH2A.X (orange) and RNAPII Ser2 (green). Scale bar represents 5 µm. The dot plot represents the relative mean intensity of the RNAPII Ser2 signal along the laser stripe compared to unaffected regions of the nucleus with error bars representing S.D. At least 22 cells were analyzed per condition from 2 or three independent experiments. A white dash line denotes the border of each nucleus. *P* values were calculated using one-way ANOVA. ****p≤0.0001, ***p≤0.001, **p≤0.01, *p≤0.05, ns = not significant.

DOI: https://doi.org/10.7554/eLife.38771.011

The following figure supplement is available for figure 4:

**Figure supplement 1.** Top, representative immunoblots corresponding to *Figure 4B*; bottom, mode of quantification of experiments shown in *Figure 4D–E*.

DOI: https://doi.org/10.7554/eLife.38771.012

same time, depletion of the FRRUC did not affect overall RAD51 protein levels as assessed by immunoblotting in both MIA-PaCa-2 and U-2OS cells (*Figure 5D* and *Figure 5—figure supplement 2A*), indicating that the complex acts by reducing the recruitment of RAD51 to chromatin rather than altering RAD51 levels. In addition, fewer cells contained BRCA1 foci after knockdown of FBXL10 or RNF68, and to a lesser extent after depletion of RNF2 (*Figure 5A–B* and *Figure 5—figure supplement 1A–B*). Silencing FBXL10, RNF68, or RNF2 also resulted in a significant reduction in the number of cells containing foci marked with phosphorylated RPA2 (pRPA2-S33), a signal associated with stretches of ssDNA (*Vassin et al., 2009*) (*Figure 5A–B* and *Figure 5—figure supplement 1A–B*). This result was confirmed using immunoblotting, which showed that phosphorylation of RPA2 on Ser4, Ser8, and Ser33 was reduced (*Figure 5D* and *Figure 5—figure supplement 2A*). Similarly, levels of phosphorylated CHK1 (Ser317) (which indicates CHK1 activation upon genotoxic stress) were lower after depletion of the FRRUC. In contrast, the levels of γH2A.X were similar in silenced cells (*Figure 5D* and *Figure 5—figure supplement 2A*), indicating that the amount of DNA damage induction was similar in all conditions. Next, using the FK2 antibody, we investigated the formation of DNA damage-induced foci containing poly-ubiquitin chains (mostly RNF8- and RNF168-dependent ubiquitylation; see Discussion), which are built in response to genotoxic stress and are critical for the DDR (*Doil et al., 2009*; *Fujimuro et al., 1994*; *Huen et al., 2007*; *Mailand et al., 2007*; *Morris and Solomon, 2004*; *Polanowska et al., 2006*; *Sobhian et al., 2007*). We observed that the depletion of the FRRUC resulted in a significant reduction in the number of cells displaying ubiquitin-positive foci (*Figure 5A–B* and *Figure 5—figure supplement 1A–B*). The percentage of cells containing γH2A.X or MDC1 foci (MDC1 recruitment is one of the most upstream events in the DDR) did not change by depleting either FBXL10, RNF68, or RNF2 (*Figure 5—figure supplement 1A–B* and *Figure 5—figure supplement 2B–C*).

These results suggest that, in response to DNA damage, the FRRUC acts in parallel with MDC1 and upstream of the formation of foci containing poly-ubiquitin chains. Furthermore, the presence of γH2A.X and MDC1 foci in all siRNA conditions indicate that the extent of DNA damage induction was similar in all conditions. Finally, we observed that RNF68 or RNF2 depletion did not influence the percentage of cells containing 53BP1, a key protein in NHEJ (*Figure 5—figure supplement 1A–B* and *Figure 5—figure supplement 2B–C*). Thus, the FRRUC seems to affect HRR, but not the NHEJ pathway. However, fewer cells contained 53BP1 foci after FBXL10 depletion, suggesting that FBXL10 may have also a role in NHEJ independently its interaction with RNF68 and RNF2, in agreement with the participation of FBXL10 (including FBXL10's isoforms not included in the FRRUC) in different protein complexes (*Sánchez et al., 2007*).

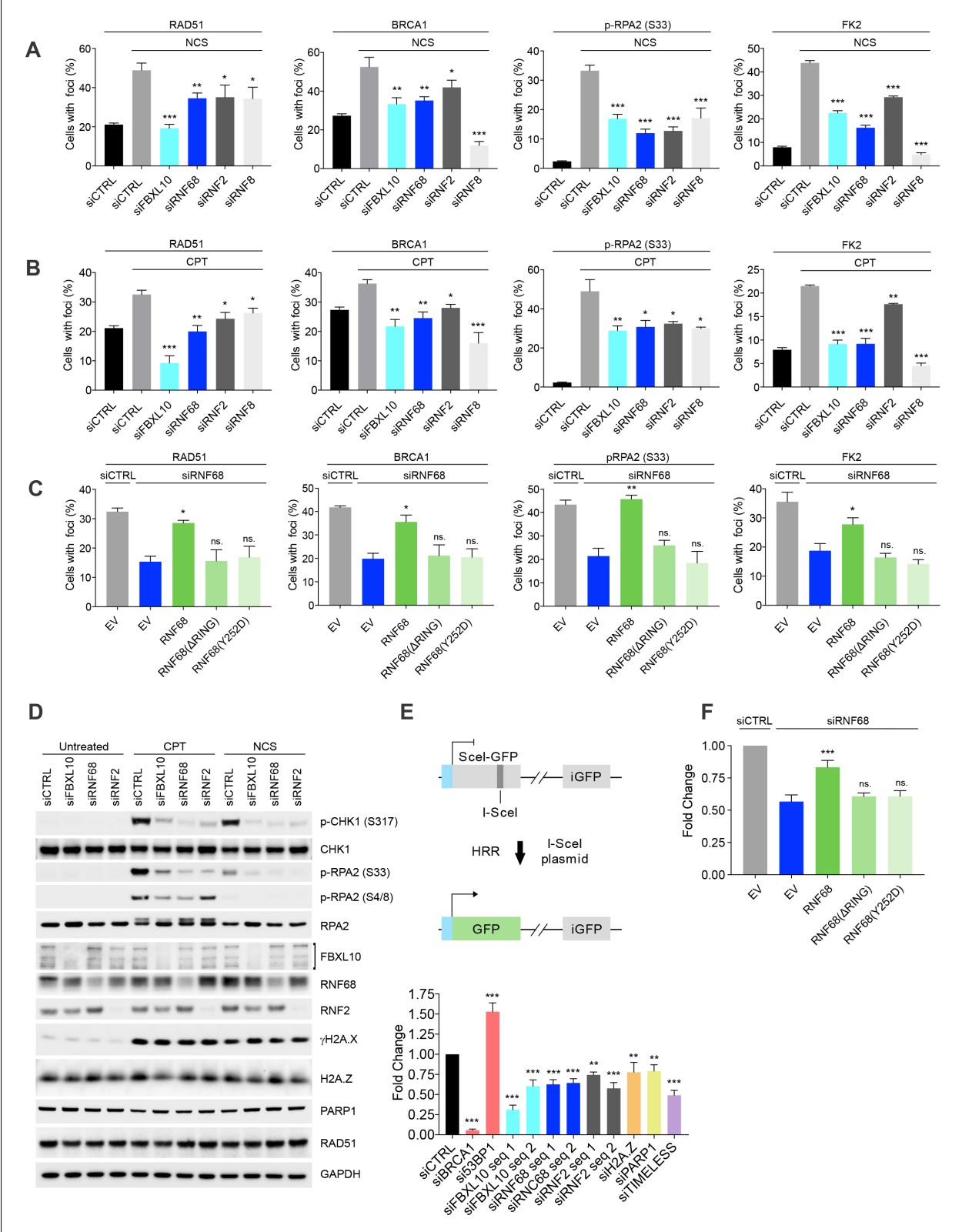

**Figure 5.** The FRUCC promotes the DNA damage response and Homologous Recombination Repair. (**A**) U-2OS cells were transfected with siRNA oligos targeting FBXL10, RNF68, RNF2, RNF8 or a non-targeting control (siCTRL). Cells were treated for 3 hr (BRCA1, Rad51) or 1 hr (FK2, pRPA S33) with 0.1 µg/mL neocarzinostatin (NCS. Cells were fixed and stained with the indicated antibodies. The percentages of cells containing DNA-damage induced foci (≥15 foci/nucleus) marked with the specific antibody are shown. At least 800 cells were counted for each condition from ≥3 independent

*Figure 5 continued on next page*

*Figure 5 continued*

experimental repetitions. Error bars represent S.E.M. Differences relative to siCTRL (NCS) were calculated using one-way ANOVA. ***$p \leq 0.001$, **$p \leq 0.01$, *$p \leq 0.05$, ns = not significant. (B) The experiment was performed in (A), except that cells were treated with 1 μM C for 6 hr (BRCA1, Rad51) or 3 hr (FK2, pRPA S33). Error bars represent S.E.M. Differences relative to siCTRL (CPT) were calculated using one-way ANOVA. ***$p \leq 0.001$, **$p \leq 0.01$, *$p \leq 0.05$, ns = not significant. (C) U-2OS cells engineered to express FLAG-HA-tagged RNF68, RNF68(ΔRING) or RNF68(Y252D) were transfected with siRNAs targeting the 3' UTR of RNF68 (siRNF68 seq2). Cells were treated with CPT for 6 hr (BRCA1, Rad51) or 3 hr (FK2, pRPA S33) and the percentages of cells containing damage-induced foci in the specified conditions were quantified. Error bars represent S.E.M from three independent experiments. *P* values were calculated using one-way ANOVA. ***$p \leq 0.001$, **$p \leq 0.01$, *$p \leq 0.05$, ns = not significant. (D) MIA-PaCa-2 cells were transfected with siRNAs targeting FBXL10, RNF68, or RNF2, or a non-targeting control (siCTRL). Cells were treated with 0.1 μg/mL NCS for 30 min or 1 μM CPT for 3 hr. Cells were lysed and immunoblotting was performed as indicated. Similar results were obtained in U-2OS cells (*Figure 5—figure supplement 2A*). (E) Upper: schematic representation of the DR-GFP-based reporter HR assay. The reporter consists of two inactive GFP genes in tandem. SceI-GFP contains a stop codon in the I-SceI endonuclease site, and iGFP is an N-terminally truncated GFP fragment. Cleavage of Sce-GFP using expression of I-SceI endonuclease introduces a double stranded break which can be repaired by HR using the downstream iGFP sequence as a template, restoring GFP expression. Lower: graph representing the quantification of HRR measured as fold change in frequency of repair of DR-GFP U-2OS cells. DR-GFP U-2OS cells were transfected with the indicated siRNAs. GFP-positive cells were measured using flow cytometry. Bars represent the mean of $\geq 4$ independent experiments ± S.E.M. *P* values were calculated using one-way ANOVA. ***$p \leq 0.001$, **$p \leq 0.01$, *$p \leq 0.05$, ns = not significant. (F) DR-GFP U-2OS were engineered to express FLAG-HA-tagged RNF68, RNF68(ΔRING) or RNF68(Y252D). Cells were transfected with siRNAs targeting the 3' UTR (siRNF68 seq2) of RNF68 and HR repair was measured as in (E). Bars represent the mean of five independent experiments ± S.E.M. *P* values were calculated using one-way ANOVA. ***$p \leq 0.001$, **$p \leq 0.01$, *$p \leq 0.05$, ns = not significant.

DOI: https://doi.org/10.7554/eLife.38771.013

The following figure supplements are available for figure 5:

**Figure supplement 1.** Representative images for *Figure 5A-B*.

DOI: https://doi.org/10.7554/eLife.38771.014

**Figure supplement 2.** The FRUCC promotes a proper DNA damage response and is needed for cell survival upon genotoxic stress.

DOI: https://doi.org/10.7554/eLife.38771.015

The stable expression of an siRNA-resistant wild type RNF68 construct in RNF68 depleted cells was able to significantly rescue the number of RAD51, BRCA1, pRPA32-S33, and FK2 foci (*Figure 5C*). In contrast, neither the siRNA-resistant RNF68(ΔRING) nor the siRNA-resistant RNF68 (Y252D) could rescue the defects in the DDR due to RNF68 depletion (*Figure 5C*). These results demonstrate that the interaction of RNF68 with both FBXL10 and RNF2, which is necessary to form an active FRRUC, is also necessary for the assembly of DNA repair foci.

Collectively, our results demonstrate that disruption of the FRRUC negatively affects the recruitment of HRR factors to sites of DNA damage and the downstream DDR.

To further assess whether this complex is required for HRR, we used the DR-GFP U-2OS reporter cells, which contain a GFP-based reporter having a recognition site for the rare-cutting endonuclease I-SceI (*Pierce et al., 1999*) (*Figure 5E*). As controls, we used BRCA1 siRNA (which is known to decrease HRR) and 53BP1 siRNA (which has been shown to promote HRR) (*Sy et al., 2009*; *Tang et al., 2013*). DR-GFP cells depleted of FBXL10, RNF68, or RNF2 exhibited significantly diminished abilities to perform HRR (*Figure 5E*). Moreover, the mislocalization of the complex by silencing of PARP1 or TIMELESS also resulted in reduced HRR capacity (*Figure 5E*), in agreement with previous reports (*Xie et al., 2015*; *Young et al., 2015*). Again, we were able to significantly rescue the effect of RNF68 depletion by stably expressing an siRNA-resistant RNF68 construct, and this rescue depended on RNF68's RING domain and ability to bind FBXL10 (*Figure 5F*). Importantly, depletion of FBXL10, RNF68, RNF2, PARP1 or TIMELESS did not have any appreciable impact on cell cycle distribution, as shown by BrdU-PI staining (*Figure 5—figure supplement 2D*), ensuring that the effects on the HHR pathway are not secondary to cell cycle differences.

No significant reduction in DNA repair mediated by NHEJ was observed upon depletion of RNF68 or RNF2, as assessed by employing a cNHEJ assay based on the EJ5-GFP reporter U-2OS cell line (*Bennardo et al., 2008*) (*Figure 5—figure supplement 2E*). Depletion of FBXL10 resulted in a reduction in NHEJ (*Figure 5—figure supplement 2E*), in agreement with the reduction in 53BP1 foci (*Figure 5—figure supplement 1A–B* and *Figure 5—figure supplement 2B–C*), again, suggesting that FBXL10 may play a role in NHEJ independently of RNF68 and RNF2.

Finally, the reduction in the DDR and HRR in cells depleted of FBXL10, RNF68, or RNF2, correlated with an increased sensitivity to DNA damage as shown by the survival curves of cells treated

with increasing concentrations of NCS (*Figure 5—figure supplement 2F*). Such hypersensitivity is suggestive of defective DNA repair.

## The FRRUC is required for the H2A/H2A.Z exchange at DNA lesions

We observed that, despite the relative increase in H2AK119Ub1 levels (*Figure 3A–C*), H2A levels were significantly reduced at the FokI nuclease-induced DNA damage sites (*Figure 6—figure supplement 1A*), in agreement with previous reports (*Batenburg et al., 2017*; *Berkovich et al., 2007*; *Strickfaden et al., 2016*). Importantly, this reduction was inhibited by the depletion of FBXL10, RNF68, or RNF2, suggesting that the FRRUC is involved in the eviction of H2A near DNA damage sites (*Figure 6—figure supplement 1A*).

H2A.Z has been shown to be rapidly incorporated into sites of DNA damage and has been implicated in making the chromatin less compact to facilitate DNA repair (*Alatwi and Downs, 2015*; *Gursoy-Yuzugullu et al., 2015*; *Xu et al., 2012*). To test whether the reduction in H2A is accompanied by a corresponding increase in the variant H2A.Z histone, we monitored the recruitment of H2A.Z to FokI-induced DSBs by qChIP 1 hr after induction, finding that the ratio of H2A.Z over H2A was increased ~4.2 and ~10.2 kbp from the DSB (*Figure 6A*). This increase was abolished after depletion of FBXL10, RNF68, or RNF2, demonstrating that the FRRUC is necessary for the enrichment of H2A. Z into sites of DNA damage (*Figure 6A*). Four hours after induction of FokI, the relative enrichment of H2A.Z over H2A was still detectable, although to a lesser extent (*Figure 6B*), in agreement with previous reports of transient incorporation of H2A.Z at DNA lesions (*Gursoy-Yuzugullu et al., 2015*; *Kalocsay et al., 2009*; *Xu et al., 2012*).

In contrast to the changes observed with H2A and H2A.Z, we detected no changes in γH2A.X accumulation at FokI-induced DSBs upon depletion of the FRRUC (at both ~4.2 and ~10.2 kbp from the break sites) (*Figure 6—figure supplement 1B*), suggesting that the amount of DNA damage induction was similar in all conditions. Importantly, in undamaged cells, the H2A.Z/H2A ratios were similar among control cells and cells depleted of either FBXL10, RNF68, or RNF2, indicating that there was no impairment in H2A.Z deposition before DNA damage induction (*Figure 6—figure supplement 1C*). We have also evaluated the enrichment of macroH2A.1, another histone H2A variant, at DSBs. We found a slight increase in macroH2A.1 levels at DSBs, but this increase was not significant (*Figure 6—figure supplement 1D*).

We wished to use a complementary technique to validate that the increase in H2A.Z levels at sites of DSBs was dependent on the FRRUC. Once again, we utilized dSTORM as an approach to precisely detect localization patterns within the nucleus. First, we confirmed that H2A.Z was enriched at DNA damage sites (*Figure 6C*), similarly to FBXL10, RNF68, and RNF2 (*Figure 1A–B*). Importantly, analysis of H2A.Z and γH2A.X in cells treated with NCS showed that the magnitude of their correlation diminished upon depletion of FBXL10, RNF68, or RNF2 (*Figure 6D* and *Figure 6—figure supplement 1E*), a further indication that the FRRUC is required for H2A.Z localization to sites of DNA damage. Accordingly, mislocalization of the FRRUC (through inhibition of PARP or depletion of PARP1 or TIMELESS) also resulted in the failure to properly localize H2A.Z at sites of DNA damage (*Figure 6D* and *Figure 6—figure supplement 1E*). To control for possible off-target effects of the siRNA, we stably expressed an siRNA-insensitive RNF68 construct and found that it was able to rescue the effect of RNF68 depletion (*Figure 6E* and *Figure 6—figure supplement 1F*). In contrast, stable expression of either an siRNA resistant RNF68(Y252D) or an siRNA resistant RNF68(ΔRING) was unable to complement the defect in H2A.Z recruitment seen in RNF68-depleted cells (*Figure 6E* and *Figure 6—figure supplement 1F*), suggesting that the ability to bind both FBXL10 and RNF2 to establish an active FRRUC is important in this process.

Combined, these measurements establish that the FRRUC is required for proper H2A.Z recruitment to sites of DNA damage.

In unperturbed human cells, the incorporation of H2A.Z-H2B dimers into nucleosomes relies on the SCRAP and P400 chromatin remodeler complexes (*Choi et al., 2009*; *Gévry et al., 2007*; *Ruhl et al., 2006*), with the P400 complex being involved in the incorporation of H2A.Z also upon DSBs (*Xu et al., 2012*). We investigated whether the shared subunits of these complexes, the AAA + helicases, TIP48 (also named RVB2) and TIP49 (also named RVB1) recruit to DNA lesions. We used laser micro-irradiation to induce local DNA damage in human U-2OS cells, which was visualized by γH2A.X staining, and detected an accumulation of TIP48 and TIP49 on the γH2A.X tracks. We found

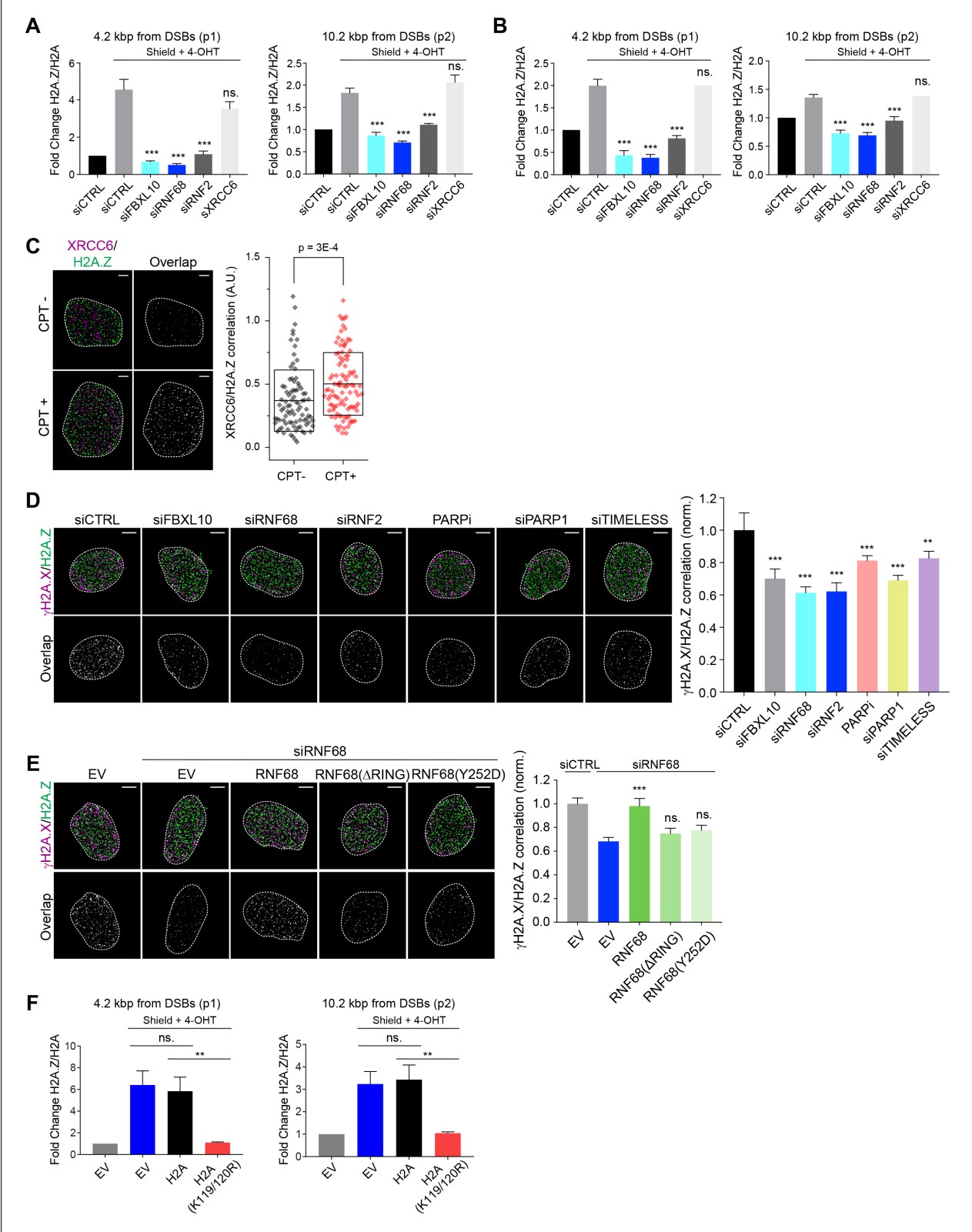

**Figure 6.** The recruitment of H2A.Z to DSBs depends on the FRUCC. (**A**) qChIP was performed in the DSB reporter U-2OS cells (see *Figure 3C*) with and without FokI-induced DSBs using the indicated antibodies. The results are shown as a ratio of H2A.Z to H2A levels. The mCherry-LacI-FokI-DD nuclease was induced for 1 hr using Shield-1 and 4-OHT. qPCR-results for amplicons 4.2 kbp (p1) and 10.2 kbp (p2) from the induced DSBs are shown. Error bars represent S.E.M from four independent experiments. *P* values were calculated using one-way ANOVA. ***p≤0.001, **p≤0.01, *p≤0.05,

*Figure 6 continued on next page*

*Figure 6 continued*

ns = not significant. (B) The qChIPs were performed as in (A) except that the mCherry-LacI-FokI-DD nuclease was induced for 4 hr using Shield-1 and 4-OHT. (C) SR imaging was performed to analyze the colocalization between XRCC6 and H2A.Z, before and after 1 hr of CPT treatment. U-2OS cells were serum starved for 72 hr, followed by release for 14 hr into complete media. Cells were fixed and stained with antibodies to XRCC6 (magenta) and H2A.Z (green). Representative nuclei are shown the left. A white dash line denotes the border of each nucleus. Scale bar represents 3 µm. The signal overlap (white) demonstrates the colocalization between XRCC6 and H2A.Z. The graph on the right displays the correlation between XRCC6 and H2A.Z in individual nuclei. Untreated, n = 87; CPT, n = 102. The number of analyzed nuclei, n, is pooled from the three independent experiments. A.U., arbitrary units. Whisker box shows mean and S.D. values. *P* values were calculated using two-sample t-test between CPT - and CPT + samples. (D) U-2OS cells were transfected with siRNAs targeting FBXL10, RNF68, or RNF2 and subjected to NCS treatment for 20 min followed by fixation and immunostaining with antibodies to γH2A.X (magenta) and H2A.Z (green). SR imaging was performed to analyze the colocalization between γH2A.X and H2A.Z. Representative nuclei are shown in the top panels, and signal overlap is shown in the lower panels. Scale bar represents 5 µm. The graph on the right represents the correlation between γH2A.X and H2A.Z after depletion of the indicated proteins. Amplitudes were normalized to the mean value of siCTRL. Error bars represent S.E.M. n = 546, 160, 172, 167, 237, 159 and 151 nuclei for siCTRL, siFBXL10, siRNF68, siRNF2, PARPi (Olaparib), siPARP1 and siTIMELESS respectively. The n value is pooled from the three independent experiments. Errors bars show S.E.M. *P* values were calculated using two-sample t-test between siCTRL and each indicated siRNA. \*\*\*p≤0.001, \*\*p≤0.01, \*p≤0.05, ns = not significant. (E) U-2OS cells stably expressing FLAG-HA-tagged RNF68, RNF68(ΔRING), RNF68(Y252D) or infected with an empty vector (EV) were transfected with siRNAs targeting the 3' UTR of RNF68 (siRNF68 seq2) or a non-targeting control (siCTRL). Cells were subjected to NCS treatment for 20 min and the extent of colocalization between γH2A.X and H2A.Z was analyzed by dSTORM microscopy, as in (D). Scale bar represents 5 µm. Correlation amplitudes were normalized to the mean value of siCTRL EV. Errors represent S.E.M. n = 363, 224, 296, 206, and 253 nuclei for siCTRL EV, siRNF68 EV, RNF68(WT), RNF68(ΔRING), and RNF68 (Y252D), respectively. The n value is pooled from three independent experiments. Differences relative to siRNA68 EV were calculated using two-sample t-test. \*\*\*p≤0.001, \*\*p≤0.01, \*p≤0.05, ns = not significant. (F) DSB reporter U-2OS cells were transiently transfected with FLAG-Strep-tagged H2A, H2A (K119/120R) or with an empty vector (EV). qChIP was performed with and without FokI-induced DSBs using the indicated antibodies. The results are shown as a ratio of H2A.Z to H2A levels. The mCherry-LacI-FokI-DD nuclease was induced for 1 hr using Shield-1 and 4-OHT. qPCR-results for amplicons 4.2 kbp (p1) and 10.2 kbp (p2) from the induced DSBs are shown. Error bars represent S.E.M from five independent experiments. *P* values were calculated using one-way ANOVA. \*\*\*p≤0.001, \*\*p≤0.01, \*p≤0.05, ns = not significant.

DOI: https://doi.org/10.7554/eLife.38771.016

The following figure supplement is available for figure 6:

**Figure supplement 1.** The recruitment to DSBs of H2A.Z, but not TIP48 and TIP49, depends on the FRUCC.

DOI: https://doi.org/10.7554/eLife.38771.017

that depletion of FBXL10, RNF68, or RNF2 did not affect the recruitment of TIP48 and TIP49 (*Figure 6—figure supplement 1G–H*).

To gain mechanistic insights on how the FRRUC and H2AK119Ub1 promote the recruitment of H2A.Z at DNA damage sites, we used a strategy previously used by the Greenberg's group (*Shanbhag et al., 2010*). Specifically, we overexpressed either wild-type H2A or H2A(K119/120R), a H2A mutant in which Lys119 and Lys120 were changed to arginines, in the inducible FokI reporter cell line. These two constructs were equally incorporated in the chromatin, as detected by ChIP, and did not affect the amount of damage to the DNA, as detected by the levels of γH2A.X at the sites of damage (*Figure 6—figure supplement 1I*). We observed a reduction in H2A.Z loading at DSBs in cells overexpressing H2A(K119/120R) compared to control cells and cells overexpressing wild-type H2A (*Figure 6F*). This result is in agreement with the fact that only wild-type RNF68, but not the two inactive RNF68 mutants, rescued the defect of H2A.Z loading (*Figure 6E*), and indicates that the FRRUC-dependent mono-ubiquitylation of H2A is required for proper H2A.Z loading.

## H2A.Z promotes local transcriptional repression, DNA damage signaling, and HRR upon DNA damage

Upon genotoxic stress, the FRRUC induces H2A mono-ubiquitylation and represses transcription at damage sites (*Figure 3* and *Figure 4*) and, at the same time, promotes H2A.Z incorporation (*Figure 6*). Therefore, we wished to investigate if H2A.Z is also required for DSB mediated transcriptional silencing. While H2A mono-ubiquitylation was not impaired upon H2A.Z silencing (*Figure 7A*), we detected significantly more nascent RNA transcript production near DSB sites in H2A.Z depleted cells compared control cells (*Figure 7B*). [H2A.Z silencing did not have a negative effect on FokI nuclease levels (*Figure 7—figure supplement 1*)]. Similarly, there was a decrease in both 5-EU incorporation and phosphorylation of RNAPII on Ser2 at laser induced damage sites in control cells, and this decrease was significantly less in cells in which H2A.Z was depleted (*Figure 7C–D*). Similar to the depletion of the FRRUC, H2A.Z depletion in both U-2OS and MIA-PaCa-2 cells also induced a

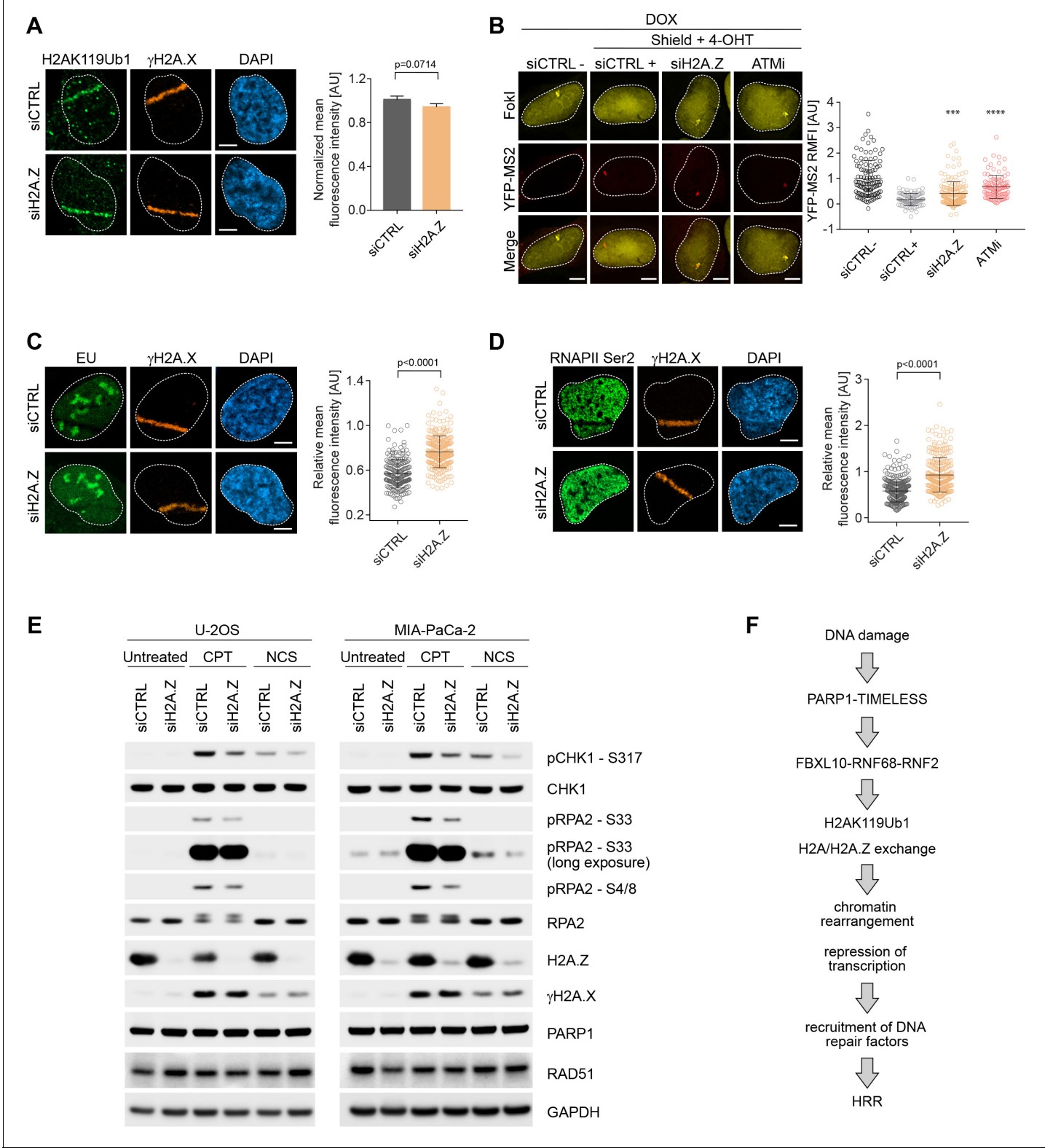

**Figure 7.** H2A.Z is involved in transcriptional repression at sites of DNA breaks and is required for efficient DSB signaling. (**A**) U-2OS cells were transfected with H2A.Z siRNA or CTRL siRNA. Cells were fixed 30 min after laser micro-irradiation and stained with antibodies to γH2A.X (orange) and H2AK119Ub1 (green). Scale bar represents 5 μm. The graph represents the mean intensity of the H2AK119Ub1 signal on the γH2A.X track, normalized to the mean values of the siCTRL sample. At least 40 laser stripes were analyzed per condition from 2 or three independent experiments. Error bars represent S.E.M. Differences relative to siCTRL were calculated using the nonparametric Mann-Whitney test. (**B**) Transcription activity was measured in

*Figure 7 continued on next page*

*Figure 7 continued*

cells silenced against H2A.Z upon induction of DSBs in U-2OS FokI-YFP-MS2 reporter cells. Quantification of YFP-MS2 relative mean fluorescence intensity (RMFI) was carried out from three independent experiments. Error bars represent S.D. n = 115, 103, 175 and 103 for siCTRL -, siCTRL +, siH2AZ and ATMi (KU60018) respectively. A white dash line denotes the border of each nucleus. Scale bar represents 5 µm. *P* values were calculated using one-way ANOVA. ****p≤0.0001, ***p≤0.001, **p≤0.01, *p≤0.05, ns = not significant. (**C**) Cells were transfected with H2A.Z siRNA or CTRL siRNA. Cells were treated with 5-Ethynyl Uridine (5-EU) immediately after laser microirradiation and were fixed 1 hr after damage induction. 5-EU was detected using Click -IT imaging kit (green) and stained with antibody to γH2A.X (orange). Scale bar represents 5 µm. The dot plot represents the relative mean intensity of the EU signal located along the laser stripe compared to unaffected regions of the nucleus with error bars representing S.D. At least 25 cells were analyzed per condition from 2 or three independent experiments. *P* values were calculated using the nonparametric Mann-Whitney test. (**D**) Right panel: Cells were stained with antibodies to γH2A.X (orange) and RNAPII Ser2 (right panel) 20 min after laser microirradiation. Scale bar represents 5 µm. The dot plot represents the relative mean intensity of the RNAPII Ser2 signal along the laser stripe compared to unaffected regions of the nucleus with error bars representing S.D. At least 25 cells were analyzed per condition from 2 or three independent experiments. *P* values were calculated using the nonparametric Mann-Whitney test. (**E**) U-2OS and MIA-PaCa-2 cells were transfected with siRNA targeting H2A.Z or a non-targeting control (siCTRL). Cells were treated with 0.1 µg/mL NCS for 30 min or 1 µM CPT for 3 hr. Cells were lysed and immunoblotting was performed as indicated. (**F**) Model of the role of the FRUCC in the response to genotoxic stress. See text for details.

DOI: https://doi.org/10.7554/eLife.38771.018

The following figure supplement is available for figure 7:

**Figure supplement 1.** FokI levels in H2A.Z depleted cells.

DOI: https://doi.org/10.7554/eLife.38771.019

reduction in the phosphorylation of RPA2 on Ser4, Ser8, and Ser33 as well as in the phosphorylation of CHK1 (Ser317) (*Figure 7E*), all events necessary for efficient HRR repair. Again, we observed no changes in γH2A.X levels upon H2A.Z silencing, showing that the amount of DNA damage induction was similar in all conditions (*Figure 7E*). Finally, we found that H2A.Z is required for efficient HRR (*Figure 5E*).

Altogether, these results indicate that H2A.Z incorporation, which is downstream to the FRUCC and H2AK119Ub1, contributes to transcriptional silencing after DNA damage, as well as efficient DNA damage signaling and HRR.

## Discussion

Epigenetic modifications and chromatin remodeling are key for modulating, propagating, and fine-tuning the DDR and associated signaling cascades necessary for DNA repair. We show here that the FRUCC is recruited to sites of DNA damage in a PARP1- and TIMELESS-dependent manner. Our results demonstrate that the complex is required for efficient mono-ubiquitylation of histone H2A on K119, a decrease of H2A levels at DSBs, localization of the histone variant H2A.Z to sites of DNA damage, transcriptional repression, DSB signaling, and proper HRR (see model in *Figure 7F*).

In response to genotoxic stress, transcriptional silencing is thought to facilitate DNA damage resolution by ensuring that the transcriptional machinery does not interfere sterically with DNA damage repair factors (e.g. by avoiding the collision of the elongating RNA Polymerase complex with DNA repair complexes) (*Uckelmann and Sixma, 2017*). Several factors and complexes, such as NurRD, PBAF, ENL, RNF8, NELF-E, and RNF51-UBR5 have been implicated in the DNA damage-induced transcriptional repression (*Awwad et al., 2017*; *Chou et al., 2010*; *Gong et al., 2017*; *Kakarougkas et al., 2014*; *Paul and Wang, 2017*; *Sanchez et al., 2016*, *2010*; *Ui et al., 2015*). One of the key determinants of transcriptional repression at sites of DNA damage is the mono-ubiquitylation of H2A on K119 (*Kakarougkas et al., 2014*; *Shanbhag et al., 2010*; *Ui et al., 2015*). We found that the mono-ubiquitylation of histone H2A on K119 is largely impaired by the depletion of the FRUCC. Similarly, the FRUCC is a major contributor to the mono-ubiquitylation of histone H2A on K119 in transcriptional regulation (*Blackledge et al., 2014*; *Rose et al., 2016*; *Taherbhoy et al., 2015*). It has been shown that RNF51-RNF2 and RNF110-RNF2 are recruited to chromatin after DNA damage to contribute to the mono-ubiquitylation of H2A on K119 (*Bergink et al., 2006*; *Ginjala et al., 2011*; *Gracheva et al., 2016*; *Ismail et al., 2010*; *Pan et al., 2011*; *Sanchez et al., 2016*). Our results show that the FRUCC is necessary for the proper recruitment of these two cPRC1s in response to genotoxic stress. Moreover, our findings suggest that the FRUCC is the most upstream ubiquitin ligase in charge of mono-ubiquitylating histone H2A on K119 and that, subsequently, cPRC1s maintain this modification. Interestingly, a recent study measured the kinetics of

recruitment of 70 DNA repair proteins to laser-induced DNA damage sites in HeLa cells (*Aleksandrov et al., 2018*). Compared to the recruitment of these 70 proteins, FBXL10, RNF68, and RNF2 display a very fast recruitment, suggesting that this represent one of the most upstream events in response to genotoxic stress.

The DNA damage response frequently re-purposes the function of specific transcriptional regulators to enact DNA repair (*Sarcinella et al., 2007*; *Shiloh and Ziv, 2013*). Our results reveal that the presence, localization, and activity of the FRRUC, in addition to mono-ubiquitylate H2A on K119 (as in unstressed cells), is also required for an efficient H2A/H2A.Z exchange at DNA lesions. Notably, whereas H2A.Z is not required for H2A mono-ubiquitylation, the FRRUC-dependent mono-ubiquity-lation of H2A is required for proper H2A.Z loading. This event, in turn, contributes to transcriptional repression, the DDR, and HRR. H2A.Z has been shown to function in the early stages of DNA repair (*Adkins et al., 2013*; *Alatwi and Downs, 2015*; *Gursoy-Yuzugullu et al., 2015*; *Jiang et al., 2015*; *Kalocsay et al., 2009*; *Xu et al., 2012*). H2A.Z is rapidly, yet transiently, recruited to DSBs (*Alatwi and Downs, 2015*; *Gursoy-Yuzugullu et al., 2015*; *Kalocsay et al., 2009*; *Kusch et al., 2004*; *Nishibuchi et al., 2014*; *Xu et al., 2012*). H2A.Z shares 56% sequence identity with H2A and the differences account for a structurally less stable interface between H2A.Z and the H3-H4 subunits of the nucleosome. In fact, H2A.Z appears to make weakened, loose nucleosomes that are easy to bend and disassemble, leading to a more accessible chromatin (*Bönisch et al., 2012*; *Jin and Fel-senfeld, 2007*; *Jin et al., 2009*; *Suto et al., 2000*; *Xu et al., 2012*), which may favor recruitment of DNA repair factors. H2A.Z has been associated with both positive and negative transcriptional regulation depending on its post-translational modifications (*Guillemette and Gaudreau, 2006*; *Hu et al., 2013*). Mono-ubiquitylation of H2A.Z on K120 and K121 is correlated with transcriptional silencing (*Draker et al., 2011*; *Sarcinella et al., 2007*), whereas H2A.Z acetylation on its N-terminus is associated with transcriptional activity (*Valdés-Mora et al., 2012*). Interestingly, mono-ubiquityla-tion of H2A.Z has been attributed to RNF2 activity (*Sarcinella et al., 2007*). Thus, the FRRUC could be involved in transcriptional silencing by several non-exclusive – and, perhaps, linked – means: (i) mono-ubiquitylating H2A, (ii) facilitating exchange of a subpopulation of H2A with H2A.Z, and (iii), mono-ubiquitylating H2A.Z. It is possible that, upon DNA damage, H2A.Z can simultaneously 'loosen' the chromatin to facilitate the access to DNA repair factors (this is the reason for which a subpopulation of H2A is exchanged for H2A.Z) and mediate transcriptional silencing (conceivably, by its mono-ubiquitylation). How mono-ubiquitylation of histones mechanistically controls at the molecular level downstream effects (such as nucleosome rearrangements, chromatin remodeling, transcriptional repression, *etc.*) remains an unsolved question in both the field of DNA damage and the field of transcription (*Hauer and Gasser, 2017*; *Nair et al., 2017*). Future investigations will be necessary to answer these fundamental questions, in general, and, more specifically, how the mono-ubiquitylation of H2A regulates its exchange with H2A.Z.

Our results indicate that the PARP1- and TIMELESS-dependent recruitment of the FRRUC to sites of DNA damage plays an early and critical regulatory role in homologous recombination repair by promoting local transcriptional inhibition both via the mono-ubiquitylation of H2A and the establish-ment of a chromatin structure containing H2A.Z. Accordingly, PARP1 has been implicated in H2A displacement and chromatin decondensation after DNA damage (*Sellou et al., 2016*; *Strickfaden et al., 2016*). The regulation of FBXL10, RNF68, and RNF2 by TIMELESS is unusual since the recruitment of other factors downstream of PARP, such as XRCC1 and Ligase 3 (that are required for single strand break and base excision repair), is affected by PARP1 silencing (*Campalans et al., 2013*; *Godon et al., 2008*; *Mortusewicz et al., 2006*; *Okano et al., 2003*), but not by TIMELESS depletion, suggesting a specific role of TIMELESS in controlling DSB repair.

In response to DNA damage, in addition to the mono-ubiquitylation of K119/K120, H2A and H2A.X are also ubiquitylated on both K13 and K15. This event requires two steps: first, an RNF ubiq-uitin ligase, namely RNF8, decorates H1 type linker histones and L3MBTL2 with K63-linked ubiquitin chains (*Mandemaker et al., 2017*; *Nowsheen et al., 2018*). Next, these chains are recognized by RNF168, another RNF ubiquitin ligase, that mono-ubiquitylates H2A and H2A.X on K13 and K15 (*Gatti et al., 2012*; *Leung et al., 2014*; *Mattiroli et al., 2012*; *Thorslund et al., 2015*). Ultimately, a K63 chain (and, possibly other kind of ubiquitin chains) is/are built on mono-ubiquitylated H2A and H2A.X, although the ubiquitin ligase involved in this elongation step is still debated (*Uckelmann and Sixma, 2017*). These ubiquitin chains are crucial for the recruitment of DNA repair factors, an event necessary for both efficient HRR and NHEJ (*Schwertman et al., 2016*). Notably, neither RNF8 nor

RNF168 can ubiquitylate K119 and K120 of H2A/H2A.X, and, vice versa, RNF2 and its interacting PCGFs are unable to ubiquitylate K13 and K15 (*Mattiroli et al., 2012*; *Wang et al., 2004*). Our findings show that, similar to the ubiquitylation of K13 and K15, the ubiquitylation of K119 and K120 in H2A/H2A.X is also involved in the recruitment of DNA repair factors. Intriguingly, while the FRRUC is necessary for an efficient DDR, ATM inhibition increases the recruitment of the FRRUC. ATM might limit the spread of the FRRUC and the subsequent H2A mono-ubiquitylation to maintain the signal at DSB proximities.

In conclusion, we show that the PARP1-TIMELESS-FRRUC axis is required for the localization of H2A.Z to sites of DNA damage. This ensures the proper transcriptional silencing and chromatin composition necessary for the efficient activation of DDR and HRR.

## Materials and methods

### Laser induced DNA damage and Live-Cell imaging

Laser-induced DNA damage induction and live-cell imaging were essentially performed as in (*Young et al., 2015*). Briefly, ~50,000 cells were plated per well of a four-well Lab-Tek II chambered number 1.5 borosilicate coverglass for 24 hr before imaging. RNAi silencing transfections were performed approximately 2 days prior to microscopy using two rounds of silencing. Cells were pre-sensitized with 10 µM BrdU (Merck) for 36 hr. For imaging, on the day of data collection, cells were incubated in FluoroBrite™ DMEM supplemented with 10% FBS, 25 mM HEPES, and sodium pyruvate. Imaging was performed using a DeltaVision Elite inverted microscope system (Applied Precision), using a 60 × oil objective (numerical aperture 1.42) from Olympus. Excitation was achieved with a 7 Color Combined Insight solid state illumination system equipped with a polychroic beam splitter with filter sets to support GFP (ex.: 475/28 nm, em.: 525/50 nm) and mCherry (ex.: 575/25 nm, em.: 632/60 nm). Images were acquired using a CoolSNAP HQ2 camera. DNA damage was generated using a 405 nm, 50 mW laser at 100% power for 0.5 s. Two pre-laser images were recorded, and the number and interval of post-laser images varied in each experiment (as indicated on the kinetic plots). Recruitment intensity was analyzed using a macro written for ImageJ (RRID:SCR_003070) that calculated the ratio of intensity of a defined laser spot *A* to the adjacent area *B* after subtraction of the background intensity of an unpopulated area of the image *C* (*Young et al., 2015*). Thus, the relative fluorescent unit (RFU) for each data collection point was calculated by the equation $RFU = (A - C)/(B - C)$. In instances where fluorescent recruitment was not detected, the coordinates of *A* were determined by use of laser coordinates recorded in the data log file. The mean values and standard errors from several cells and live-cell imaging time courses were computed for each time point using Origin (2016) software. To visualize the recruitment of endogenous FBXL10, RNF68 and RNF2, laser induced laser stripes were done on a Zeiss LSM 800 microscope (RRID:SCR_015963), using a 405 nm diode laser (5 mW) with the timed bleach option (60 iterations, 80% laser power output) in the ZenBlue 2.1 software using a Plan-Apochromat 63x/1.40 Oil objective after pre-sensitizing the cells with 1 µg/ml Hoechst 33342 for 30 min.

### Plasmids

FBXL10, RNF68, RNF2, and FBXL11 cDNAs were inserted into several different vector backbones, including modified pcDNA3.1 vectors containing N-terminal FLAG/HA/myc-tags, pBABE.puro retroviral vectors containing N-terminal FLAG/HA/myc/GFP-tags; and pRetroQ-mCherry-C1 retroviral vector containing an mCherry-tag. Site-directed mutagenesis was performed using KAPA polymerase (KAPA Biosystems). Specific details will be provided on request.

### Antibodies

The following antibodies were used: BRCA1 (1:200 (IF), 1:1000 (WB), Bethyl cat. No. A301-377A, RRID:AB_937735), BrdU (1:100, DAKO cat. No. M0744, RRID:AB_10013660), pCHK1 (S317) (1:1000, Cell Signaling cat. No. 2344S, RRID:AB_331488), CHK1 (1:1000, Santa Cruz Biotechnology cat. No. sc-8408, RRID:AB_627257), c-myc (1:2000, Bethyl cat. No. A190-105A, RRID:AB_67390), FLAG (1:2000, Sigma-Aldrich cat No. F1804, RRID:AB_262044), FK2 (1:1000, Merck cat. No. 04–263, RRID:AB_612093), GAPDH (1:5000, Sigma-Aldrich cat No. G8795, RRID:AB_1078991), HA.11 (16B12) (1:5000, Covance cat. No. MMS-101P, RRID:AB_2314672), HA (1:2000, Bethyl cat. No. A190-108A,

RRID:AB_67465), MDC1 (1:2000, Bethyl cat No. A300-051A, RRID:AB_203282), Mouse Gamma Globulin Control (10 μg (ChIP), ThermoFisher Scientific cat. No. 31878, RRID:AB_2532170), Rabbit Gamma Globulin Control (10 μg (ChIP), ThermoFisher Scientific cat. No. 31887, RRID:AB_2532177), H2A (1:1500 (WB) 10 μL (ChIP), Active Motif cat. No. 39111), H2AK119Ub1 (1:1000 (WB), 1:1000 (IF), 10 μL (ChIP), EMD Millipore cat. No. 05–678, RRID:AB_309899), γH2A.X (1:1500 (WB), 1:5000 (IF), 5 μL (ChIP), EMD Millipore cat. No. 05–636, RRID:AB_309864), γH2A.X (1:1000 (IF), Cell Signaling Technology cat. No. 9718, RRID:AB_2118009), H2A.Z (1:1000 (WB), 1:2000 (IF), 5 μL (ChIP), Active Motif cat. No. 39113, RRID:AB_2615081), Histone H3 (1:30000 (WB), 3 μL (ChIP), Abcam cat. No. ab1791, RRID:AB_302613), mCherry (1:1000, LifeSpan BioSciences cat. No. LS-C204826), macroH2A.1 (2 μL (ChIP), Abcam cat No. ab37264, RRID:AB_883064), RAD51 (1:500 (IF), GeneTex cat. No. GTX70230, RRID:AB_372856), RAD51 (1:1000 (WB), Santa Cruz Biotechnology cat. No. sc-8349, RRID:AB_2253533), PARP1 (1:1000, Cell Signaling Technology cat. No. 9542S, RRID:AB_2160739), RNA Polymerase II phospho-Ser2 (1:1000 (IF), BioLegend cat. No. 920204, RRID:AB_2616695), RNF2 (1:1000 (WB and IF), Cell Signaling Technology cat. No. 5694S, RRID:AB_10705604), RNF51 (1:1000 (WB), Bethyl cat No. A301-694A, RRID:AB_1210891), RNF68 (1:1000 (WB and IF), Sigma-Aldrich cat. No. HPA011356, RRID:AB_1855063), RNF68 (1:1000 (WB) Santa Cruz Biotechnology cat. No. sc-515371, RRID:AB_2721914), RNF110 (1:1000 (WB) Millipore cat No. 09–724, RRID:AB_11212898), RPA32 (1:5000, Merck cat. No. MABE285, RRID:AB_11205561), pRPA32 (S4/S8) (1:5000 (WB), Bethyl cat No. A300-245A, RRID:AB_210547), pRPA32 (S33) (1:2000 (IF), 1:5000 (WB), Bethyl cat No. A300-246A, RRID:AB_2180847), TIMELESS (1:1000, Bethyl cat. No. A300-961A, RRID:AB_805855), TIP48 (1:100 (IF), Proteintech, cat No. 10195–1-AP, RRID:AB_2184679), TIP49 (1:500 (IF), Millipore cat No. 06–1299, RRID:AB_10806083), TP53BP1 (1:1000 (IF), Abcam cat. No. ab36823, RRID:AB_722497), αTubulin (1:5000 (WB), Sigma-Aldrich cat No. T6074, RRID:AB_477582), XRCC5 (1:1000 (IF), Abcam cat. No. ab198587), XRCC6 (1:1000 (WB), Santa Cruz Biotechnology cat. No. sc-9033, RRID:AB_650476), XRCC6 (1:2000 (IF), ThermoFisher Scientific cat. No. MA5-13110, RRID:AB_10976973). A custom affinity-purified rabbit polyclonal antibody to FBXL10 was produced by Genscript (RRID: SCR_002891) against the following peptide sequence: N-CNHAGKTGKQKRGPG-C). Antibodies to SKP1 have been described previously (Pagan et al., 2015).

## Cell culture, transfection and drug treatment procedures

Cell lines were purchased from ATCC and were routinely checked for mycoplasma contamination with the Universal Mycoplasma Detection Kit (ATCC 30–1012K). HEK293T (ATCC CRL-3216, RRID: CVCL_0063) and MIA-PaCa-2 (ATCC CRL-1420, RRID:CVCL_0428) cells were maintained in Dulbecco's modified Eagle's medium (Gibco). U-2OS (ATCC HTB-96, RRID:CVCL_0042) cells were maintained in McCoy's 5A medium (Gibco). Media was supplemented with 10% FBS (Corning Life Sciences) and 1% penicillin/streptomycin/L-glutamine (Corning Life Sciences). All cell lines were maintained at 37°C at 5% $CO_2$ in a humidified atmosphere. Cells were treated with 0.1 μg/ml neocarzinostatin (Sigma) or 1 μM CPT (Sigma). Cells were pretreated with inhibitors to ATM (KU60019, Tocris, 5 μM), ATR (AZ20, Tocris, 1 μM), DNA-PK (NU7441, Tocris, 5 μM), and PARP (PJ34, EMD Millipore, 300 nM or Olaparib, Selleck Chemicals, 1 μM) for 1 hr prior to laser-induced DNA damage. Both U-2OS reporter cell lines having the inducible ER-mCherry-LacI-FokI-DD nuclease used for qChIP and to measure transcription were a gift from Prof. Roger A. Greenberg (University of Pennsylvania) (Tang et al., 2013). ER-mCherry-LacI-FokI-DD was induced with Shield1 ligand (250 nM) and 4-OHT (500 nM) for the indicated amount of time.

U-2OS cells were transiently transfected using Lipofectamine 3000 (ThermoFisher Scientific) based on the manufacturer's recommendation. HEK293T cells were transiently transfected using polyethylenimine (Pagan et al., 2015; Skaar et al., 2015). For retrovirus production, HEK293T cells were transfected with pBABE.puro or pRetro-Q backbone vectors containing the genes of interest, in combination with Gag-Pol and VSV-G plasmids (Clontech) (Dankert et al., 2016). siRNA oligo transfections were performed using RNAiMax (ThermoFisher Scientific) according to the manufacturer's instructions.

The following siRNA duplexes from Dharmacon were used: D-001810–01 (non-targeting control); L-014930–00 (FBXL10 seq1); CCUUAAGGUAGGAGAGAAUdTdT (FBXL10 seq2); L-007094–00 (RNF68 seq1); CCCAGAUAUUUAUGUGAAAUU (RNF68 seq2); L-006556–00 (RNF2 seq1); UUAAAGAUGUACUGGCAUUUU (RNF2 seq2); L-005230–01 (RNF51); L-006584–00 (RNF110);

L-003461–00 (BRCA1); L-003548–00 (TP53BP1); L-006900–00 (RNF8); L-005084–00 (XRCC6); L-006659–06 (PARP1); L-019488–06 (TIMELESS); L-011683–00 (H2AFZ).

## Immunoprecipitation and immunoblotting

Cells were lysed in lysis buffer (25 mM Tris pH 8.0, 150 mM NaCl, 10% glycerol, 1 mM EDTA, 1 mM EGTA, 1 mM 1,4-Dithiothreitol (DTT) and 0.1% NP-40) supplemented with protease (Complete ULTRA, Roche) and phosphatase inhibitors (PhosSTOP, Roche). The insoluble fraction was removed by centrifugation (20,000 x g x 15 min at 4°C). Immunoprecipitations and affinity precipitations were carried out using FLAG-M2 agarose (Sigma-Aldrich) or anti-HA Affinity Matrix (Roche) at 4°C for 1 hr. Beads were washed once in lysis buffer before washing with CSK buffer (10 mM PIPES, pH 7.0, 100 mM NaCl, 300 mM sucrose, 1 mM EGTA, 3 mM $MgCl_2$, 0.1% Triton X-100) containing 1 U/mL TurboNuclease (Accelagen) for 15 min at RT. Beads were extensively washed in lysis buffer and elution was carried out with 3 × FLAG peptide (Sigma-Aldrich) or HA peptide (Roche). Immunoblotting was performed as previously described (*Pagan et al., 2015*).

## Immunofluorescence microscopy

Cells cultured on coverslips were either pre-extracted with ice-cold CSK buffer containing 0.5% Triton X-100 for 5 min, or directly fixed in 4% paraformaldehyde/PBS after a PBS wash. For endogenous FBXL10, RNF68 or RNF2 staining, samples were heated to 65°C for 10 min in an epitope unmasking buffer (10 mM Sodium citrate, 0.05% Tween 20, pH 6.0). Samples were then blocked in blocking buffer (PBS/0.05% Triton X-100 containing 5% FBS and 3% BSA) before incubation with indicated primary antibodies in blocking buffer. Alexa-555 and Alexa-488-conjugated secondary antibodies were from Life Technologies. Slides were mounted in ProlongDiamond with DAPI (Molecular probes). Imaging was performed using a DeltaVision Elite inverted microscope system (Applied Precision), using a 60x objective (numerical aperture 1.42) from Olympus or a Zeiss LSM 800 microscope using a Plan-Apochromat 63x/1.40 Oil objective.

## Nascent transcript detection at DNA damage sites

DSBs in the U-2OS reporter cells were induced by activating the ER-mCherry-LacI-FokI-DD with Shield-1 ligand (250 nM) and 4-OHT (500 nM) for 4 hr. Transcription was induced from the integrated reporter gene by treating the cells with 1 µg/ml doxycycline (DOX) for 3 hr. Where indicated, ATM inhibitor was applied 30 min before the addition of DOX. For 5-EU and RNAPII Ser2 staining, U-2OS cells were subjected to laser-microirradiation. 5-ethynyl uridine (5-EU) was added to the media immediately after laser treatment to a final concentration of 1 mM followed by incubation at 37°C for 1 hr. Incorporation of 5-EU was detected by Click-iT RNA imaging kit (Invitrogen) following manufacturer's instructions. Samples were further processed for γH2A.X staining. Alternatively, laser treated samples were fixed 20 min after damage induction and stained with an antibody against RNAPII (phospho-Serine 2) or γH2A.X. Relative fluorescence values were calculated by measuring 5-EU or RNAPII Ser2 signal intensities along the laser track highlighted by the γH2A.X staining and compared with undamaged regions of the nucleus using ImageJ as shown in *Figure 4—figure supplement 1B*.

## Cell cycle analysis

BrDU incorporation and Propodium Iodide staining was performed essentially as described in (*Clijsters et al., 2013*). Briefly, U-2OS cells were pulse-labelled with 10 µM BrdU (Merck) for 30 min followed by trypsinization and overnight fixation in ice-cold 70% ethanol. After washing with PBS and then with 1% BSA, 0.05% Triton X-100 in PBS (PBST-BSA), DNA was denatured for 20 min with 2 N HCl, 0.5% Triton X-100. Cells were washed with PBS, and resuspended in sodium tetraborate (0.1 M, pH 8.5) for 5 min. Cells were incubated with anti-BrdU antibody (1:100, DAKO) in PBST-BSA for 1 hr at RT. After washing, Alexa Fluor-488 coupled goat anti-mouse (1:500, Molecular Probes) secondary antibody was added in PBST-BSA for 30 min at RT. After several washes, cells were resuspended in PBST-BSA with propidium iodide (10 µg/mL, Invitrogen) and RNase A (20 µg/mL, Invitrogen). Cell cycle analysis was carried out by flow cytometry with a BD FACSCalibur Cell Analyzer and FlowJo (RRID:SCR_008520) software.

## HRR and NHEJ assays

I-SceI-based reporter assays were used to measure the efficiency of repair by HR and NHEJ pathways (*Bennardo et al., 2008*; *Gunn and Stark, 2012*; *Pierce et al., 1999*). DR-GFP-U-2OS cells, from The M. Jasin Lab, were used for HR assays, and EJ5-U-2OS cells, from The J. Stark Lab, were used for NHEJ assays. Cells were grown in six well dishes and were transfected with siRNA oligos using RNAiMax (ThermoFisher Scientific) according to manufacturer's specifications. The following day, cells were transfected with 1 μg of the I-SceI-GR-RFP plasmid (a gift from Tom Misteli, Addgene # 17654, RRID:SCR_002037) using Lipofectamine 3000. Cells were subjected to a second round of siRNA silencing. To induce I-SceI-GR-RFP expression, cells were treated with 0.2 μM triamcinolone acetonide (TCA, Sigma-Aldrich) for 48 hr. Flow cytometry analysis was performed using a BD LSR II HTS Analyzer (RRID:SCR_002159). The percentages of RFP positive cells that were also GFP positive were quantified.

## Chromatin immunoprecipitation (ChIP)

ChIP was performed essentially as described in (*Busino et al., 2012*). Cells were cross-linked with 1% formaldehyde for 10 min, then quenched with the addition of 0.2 M Glycine. Lysates were sonicated in a Bioruptor XL (Diagenode) to yield DNA fragments ~500 bp in size. Immunoprecipitations were performed overnight at 4°C using the specified antibodies. IgGs were used as a negative control. Precipitates were washed sequentially with buffer I (0.1% SDS, 1% Triton X-100, 2 mM EDTA, 20 mM Tris–HCl, at pH 8.1, and 150 mM NaCl), buffer II (0.1% SDS, 1% Triton X-100, 2 mM EDTA, 20 mM Tris–HCl, at pH 8.1, and 500 mM NaCl) and buffer III (0.25 M LiCl, 1% NP-40, 1% deoxycholate, 1 mM EDTA and 10 mM Tris–HCl, at pH 8.1). Precipitates were then washed three times with TE buffer and eluted three times with 1% SDS and 0.1 M NaHCO$_3$. Eluates were pooled and heated at 65°C for at least 6 hr to reverse the formaldehyde crosslinking. DNA fragments were precipitated with a standard phenol/chloroform protocol. Quantitative PCR analysis was performed using ABsolute SYBR green master mix (Thermo Scientific) and a Roche Lightcycler 480. Each primer set gave a single and specific product when assessed using PCR dissociation curves. Results were quantified relative to input DNA. ChIP qPCR data were expressed either as percentage of input or fold enrichment relative to untreated samples. qChIP primer sequences are as follows:

~4.2 kbp distance from the lacO repeats (p1: F: 5'-TGTACGGTGGGAGGCCTATATAA-3', R: 5'-GCGTCTCCAGGCGATCTG-3'); ~10.2 kbp distance from the lacO repeats (p2: 5'-CCACCTGACGTC TAAGAAACCAT-3', R: 5'-GATCCCTCGAGGACGAAAGG-3'). qRT-PCR

Total RNA was generated using RNeasy mini kits (Qiagen). cDNA was generated using Random Hexamer or OligodT EcoDry kits (Takara Clontech). qPCR was performed using Absolute SYBR green (Thermo Fisher Scientific) on a Roche Lightcycler 480 PCR instrument. For each biological sample, triplicate reactions were analyzed using absolute relative quantification method alongside in-experiment standard curves for each primer set to control for primer efficiency. The oligos used for qRT-PCR analysis were: FBXL10 (F: 5'-CGGCAAGACCGGGAAACAAAAG-3', R: 5'-TGTCCCGG TTCATCTTCTGC-3'); RNF68 (F: 5'-GTGTACAAGATGGACCCGCT-3', R: 5'- ACTCTTGCAGAAAGTA TGAAGACAC-3'); RNF2 (F: 5'-AGACAAACGGAACTCAACCA-3', R: 5'-ATTGCCTCCTGAGGTG TTCG-3'); RNF8 (F: 5'-GTGCGAGGTGACTGTAGGAC-3', R: 5'-CGAGAAATCATCAGGGGGCA-3'); PARP1 (F: 5'-GCCCTAAAGGCTCAGAACGA-3', R: 5'-CTACTCGGTCCAAGATCGCC-3'), TIMELESS (F: 5'-TGCTGGAGCCTACAAAGAGC-3', R: 5'-GATGAGCCTCTTCTTCGGGG-3'), H2A.Z (F: 5'-AGCGGAATTCGAAATGGCTG-3', R: 5'-GAATACGGCCCACTGGGAAC-3' GAPDH (F: 5'-TGCAC-CACCAACTGCTTAGC-3', R: 5'-GGCATGGACTGTGGTCATGAG-3'), XRCC6 (F: 5'-CATGGCAAC TCCAGAGCA-3', R: 5'-GCTCCTTAAACTCATCCACC-3'), RNF51 (F: 5'-TGGTTGCCCATTGA-CAGCG-3', R: 5'-ACACGTTTTACAGAAGGAATGTAGA-3'), RNF110 (F: 5'-CCGAACCCAGA TGGCCGAAA-3', R: 5'-GTTGCCCTCGGAACAGGGTC-3')

## Viability assay

MIA-PaCa-2 cells were silenced with the indicated siRNAs for 2 rounds for 72 hr. Cells were then plated in triplicates at densities of 2000 cells per 96 well plate and incubated for 24 hr before exposure to various doses of NCS. After 72 hr, AlamarBlue assay (BioRad) was used to measure the metabolic activity of cells according to the manufacturer's instructions.

## Statistical analysis

Data were analyzed using GraphPad Prism 7.03 software (RRID:SCR_002798). Two-group datasets were analyzed using the nonparametric Mann-Whitney test unless otherwise stated. For comparisons between three or more groups, parametric or non-parametric one-way analysis of variance (ANOVA) was used.

## dSTORM imaging

Samples were prepared as in *Immunofluorescence microscopy* using Alexa Fluor-568 and Alexa Fluor-647 as secondary antibodies. The dSTORM imaging was performed on a custom-built optical imaging platform based on an inverted microscope (Leica DMA 300) (*Chen et al., 2015*). The microscope was equipped with 561 nm (200 mW, UltraLasers, MGL-FN-561–200), and 640 nm (150 mW, OEM Laser systems, MLL-III-150) laser lines, as well as a CORR total reflection objective (Zeiss, 100X, NA = 1.47, HCX PL APO). The two laser lines were collimated and reflected by a penta-edge dichroic beam splitter (Semrock, 408/504/581/667/762-Di01−22 × 29). Cell samples were excited in Highly inclined and Laminated Optical sheet (HILO) illumination mode (*Tokunaga et al., 2008*). A 405 nm laser line (100 mW, Applied Scientific Pro., MDL-III-100) was employed to enhance photo-switching of fluorophores. For two-color dSTORM imaging, Alexa Fluor-568 and Alexa Fluor-647 were sequentially illuminated and their emitted fluorescence were sequentially collected by a (a bandpass filter 676/37 (Semrock) for Alexa Fluor 647, and a bandpass filter 607/36 (Semrock) for Alexa Fluor 568. The photons were recorded by a sCMOS camera (Prime 95B, Photometrics). The two colors were aligned by matching the wide-spectrum fluorescent beads (diameter = 100 nm, TetraSpec, T7279) via a third order polynomial algorithm. The two single-band pass filters for different fluorophores were switched via a filter wheel (FW102C, Thorlabs). 2000 frames per image for both colors were recorded at a frequency of 33 Hz. The imaging buffer, composed of an oxygen scavenging system (1 mg/mL glucose oxidase, 0.02 mg/mL catalase, and 10% glucose in PBS) and 100 mM cycteamine (MEA), was freshly mixed before injection onto the sample coverslips.

## Image reconstruction

Single-molecule image stacks were reconstructed by fitting each single-molecule event into a 2D Gaussian Point Spread Function (PSF) and recording the successively fitted central coordinates of the PSFs. Briefly, each 2D Gaussian PSF was fitted using the Maximum-Likelihood Estimation algorithm, with the characteristics of each individual camera pixel (modeled as a Gaussian distribution) considered (*Huang et al., 2013*). Each fitting was evaluated by its Log-Likelihood Ratio (LLR) against the null-hypothesis, and the insignificant fittings (at 0.05 level) were discarded. Each fitting accuracy (precision) was given by its Cramér-Rao Lower Bound (CRLB). The frequency of each fitting accuracy was then counted, and its histogram was fitted into a skewed Gaussian function, of which the center was given as an average fitting accuracy. In this work, the fitting accuracy of Alexa Fluor-647 is ~8 nm, and that of Alexa Fluor-568 is ~10 nm. The recorded coordinates were then rendered into images of 10 nm/pixel or 5 nm/pixel for display or correlation analysis purposes, respectively.

## Cross-Pair-Correlation analysis

Briefly, fitted coordinates of each PSF center were rendered into two-color images of 5 nm/pixel. An $8 \times 8 \ \mu m^2$ square was then cropped from the center of one nucleus image. The two colors of the square were then split into separate binary images (2D arrays) for cross-pair-correlation analysis.

Cross-Pair-Correlation was defined as follows (*Veatch et al., 2012*):

$$g(\mathrm{r}) = \frac{\langle \rho_1(R)\rho_2 \, (\mathrm{R}+\mathrm{r})\rangle_{\mathrm{R}}}{\langle \rho_1(R)\rangle_{\mathrm{R}}\langle \rho_2 \, (\mathrm{R})\rangle_{\mathrm{R}}}$$

where $\langle K_{\mathrm{R}}\rangle$ denotes the ensemble average across each pixel $R$, and $\rho_x(R) = 1$ (or $\rho_x(R) = 0$) describes whether there is (or not) a single-molecule event detected in the $x$th Channel at pixel $R$. $\rho_x = \langle\rho_x(R)\rangle_{\mathrm{R}} = N_x/S.$ denotes the averaged density of all the single-molecule events, where $N_x$ is the number of single-molecule events on the canvas of size $S$. In practice, the correlation function was calculated based on the Fourier Transform theorems and the Matlab (MathWorks Inc.) built-in 2D Fast-Fourier-Transform (FFT) functions (fft2, ifft2). The calculated correlation function was then transferred from cartesian to polar coordinates, and radially averaged through $[0, 2\pi]$. It was then

fitted into a 1D Gaussian function, and the amplitude was read out as the descriptive correlation strength. We note that the correlation length (distance) describes a spatial range within which the correlation between the two species becomes significant as compared to the random colocalizations. It cannot be considered as a measure of the absolute center-to-center distance between two molecules. The pair-cross-correlation was performed to robustly map the colocalization between tested samples. We note that this pair-cross-correlation collects the pairwise distances between two species and computes their probability density in the spatial frequency domain. The false positive (random) colocalization events that are caused by the sample's dense distributions in the nucleus, therefore, only contribute as the denominator, so that the pair-cross-correlation demonstrates the relative amount of significant (true positive) colocalization events versus the overall (true positive + false positive) colocalization observations. Representatives images were smoothed, thresholded, and binarized before they were overlapped. We note that these post-imaging procedures are for display purpose. The correlation analysis was performed directly on the reconstructed raw coordinates and did not include these post-processing artifacts.

## Acknowledgements

The authors thank R Greenberg, M Jasin, and J Stark for reagents; and D Durocher, R Greenberg, P Lee, J Lukas, Y Shiloh, and T Sixma for critically reading the manuscript. Authors are grateful to Chengbo Chen for the ImageJ scripts. MP is grateful to TM Thor and DR is greatful to S Perrotta for continuous support. This work was funded by a Rosztoczy Foundation Fellowship to GR, grants from the National Institutes of Health (R01-CA076584 and R01-GM057587) to MP, (R01-GM057691, R21-CA187612) to ER and the American Cancer Society grant (ACS 130304-RSG-16-241-01-DMC) to ER.

## Additional information

### Funding

| Funder | Grant reference number | Author |
| --- | --- | --- |
| The Rosztoczy Foundation | Fellowship | Gergely Rona |
| American Cancer Society | ACS 130304-RSG-16-241-01-DMC | Eli Rothenberg |
| National Institutes of Health | R01- GM057691 | Eli Rothenberg |
| National Institutes of Health | R21-CA187612 | Eli Rothenberg |
| National Institutes of Health | R01-CA076584 | Michele Pagano |
| National Institutes of Health | R01-GM057587 | Michele Pagano |
| Howard Hughes Medical Institute | | Michele Pagano |

The funders had no role in study design, data collection and interpretation, or the decision to submit the work for publication.

### Author contributions

Gergely Rona, Domenico Roberti, Conceptualization, Data curation, Formal analysis, Supervision, Validation, Investigation, Visualization, Methodology, Writing—original draft, Writing—review and editing; Yandong Yin, Formal analysis, Investigation, Methodology, Writing—original draft; Julia K Pagan, Conceptualization, Data curation, Supervision, Investigation, Writing—original draft, Writing—review and editing; Harrison Homer, Elizabeth Sassani, Formal analysis, Investigation; Andras Zeke, Conceptualization, Formal analysis; Luca Busino, Conceptualization, Data curation, Formal analysis, Investigation, Writing—original draft, Writing—review and editing; Eli Rothenberg, Conceptualization, Data curation, Formal analysis, Supervision, Funding acquisition, Validation, Investigation, Methodology, Writing—original draft, Writing—review and editing; Michele Pagano, Conceptualization, Resources, Data curation, Formal analysis, Supervision, Funding acquisition, Investigation, Methodology, Writing—original draft, Writing—review and editing

## Author ORCIDs

Gergely Rona (iD) http://orcid.org/0000-0003-3222-7261
Yandong Yin (iD) http://orcid.org/0000-0003-2499-871X
Harrison Homer (iD) http://orcid.org/0000-0002-7157-0196
Luca Busino (iD) https://orcid.org/0000-0001-6758-9276
Eli Rothenberg (iD) http://orcid.org/0000-0002-1382-1380
Michele Pagano (iD) http://orcid.org/0000-0003-3210-2442

## Decision letter and Author response

Decision letter https://doi.org/10.7554/eLife.38771.025
Author response https://doi.org/10.7554/eLife.38771.026

## Additional files

### Supplementary files

• Source data 1. Raw data for all graphs in main figures and figure supplements.
DOI: https://doi.org/10.7554/eLife.38771.020

• Transparent reporting form
DOI: https://doi.org/10.7554/eLife.38771.021

### Data availability

All data generated or analyzed during this study are included in the manuscript and supporting files. Source data files have been provided for: Figure 1B-C, Figure 2A-C, Figure 3A-C, Figure 4B and D-E, Figure 5A-C and E-F, Figure 6A-F, Figure 7A-D, Fig.Sup. 2C-D, Fig.Sup. 3B-C, Fig.Sup. 4A, Fig. Sup. 6B-C and E-F, Fig.Sup. 7A-I

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
