## [Decision Letter]

Thank you for submitting your work entitled "PARP1-dependent recruitment of the FBXL10-RNF68-RNF2 ubiquitin ligase to sites of DNA damage controls H2A.Z loading" for consideration by *eLife*. Your article has been reviewed by three peer reviewers, one of whom is a member of our Board of Reviewing Editors, and the evaluation has been overseen by a Senior Editor. The reviewers have opted to remain anonymous.

Our decision has been reached after consultation between the reviewers. Based on these discussions and the individual reviews below, we regret to inform you that your work will not be considered further for publication in *eLife*. However, we agree that this is a paper of interest for *eLife* in the future if the main concerns raised are appropriately responded to with additional experiments.

This manuscript describes a series of complex and high-quality experiments using a varied combination of cell biology and molecular biology approaches to conclude that the core ubiquitin ligase subcomplex of the ncPRC1.1 Polycomb repressive complex formed by FBXL10, RNF2 and RNF68 is recruited to DNA double strand breaks in a PARP1 and TIMELESS dependent manner. Recruitment is accompanied by mono-ubiquitylation of H2A K119, consistent with the activity of the FBXL10-RNF2-RNF68 complex, and a reduction of H2A density that is accompanied by an increase of H2AZ. As a consequence of these changes, transcription is reduced at the site of damage. Finally, the authors show that FBXL10-RNF2-RNF68 is required for efficient homologous recombination repair of the DSB. The study thus provides a general picture of the control of HR DSB repair by FBXL10-RNF68-RNF2 complex in a PARP1 and TIMELESS-dependent manner, but a mechanistic insight of how this complex promotes H2AZ incorporation is lacking. Despite the potential and interest of the work for *eLife*, the H2A.Z part, critical for the novelty of the manuscript, should be developed further. It would be essential to figure out the connection between ncPRC and cPRC and H2AZ deposition. Additionally, the STORM data does not seem sufficiently valid as it is and needs additional work to show the claims. More specific comments can be found in the individual reviews attached below.

Reviewer #1:

This is a nice manuscript describing a series of complex and high-quality experiments using a varied combination of cell biology and molecular biology approaches to conclude that the core ubiquitin ligase subcomplex of the ncPRC1.1 Polycomb repressive complex formed by FBXL10, RNF2 and RNF68 is recruited to DNA double strand breaks in a PARP1 and TIMELESS dependent manner. Using the powerful technology of STORM, the authors first show using neocarzinostatin to generate breaks a rapid recruitment to damages and confirm this result using a known fusion construct of the Fok1 endonuclease that cleaves specifically to a lacO repeated region of a modified U2OS line. This action is specific for this complex as the FBXL11 paralog of FBXL10, which is not part of the ncPRC1.1 complex, does not recruit to the damage. Authors show that recruitment is dependent on PARylation by using the PARP1 inhibitor Olaparib and by knocking down PARP1 and TIMELESS in the cells. The reaction is accompanied by mono-ubiquitylation of H2A K119, consistent with the activity of the complex, and a reduction of H2A density that is accompanied by an increase of H2AZ. As a consequence of these changes, authors are able to show that transcription is reduced at the site of damage as determined by 5-EU global labeling of RNA as well as specific system based on a reporter containing MS2stem loop repeats to visualize transcription directly at the transcription site. Concomitantly they see a lower Ser2P RNAPII, consistent with lower transcription elongation efficiency. Analysis of the impact of depleting a number of homologous recombination and DSB-signals (RAD51, BRCA1, CHK1 phosphorylation) authors provide evidence that depletion of FBXL10-RNF2-RNF68 reduces homologous recombination. Finally the authors identify the minimal region required of four amino acids of RNF68 required for stable binding with FBXL10 and show by generating mutants like RNF68Y252D, which reduces binding to FBXL10 but not to RNF2, and mutants deleting the RING domain of RNF68, which did not affect FBXL10 binding but it did RNF2 binding, that disruption of the complex affects recruitment of recombination factors to the breaks. Recombination was assayed with the I-SceI DSB system of M. Jasin to corroborate functional meaning of the results, providing evidence that this is at least in part due to the replacement of H2A by H2AZ at the site of the DSB promoted by the FBXL10-RNF68-RNF2 complex. Therefore, the manuscript provides an important regulatory step of DSB repair via homologous recombination that is controlled by the FBXL10-RNF68-RNF2 complex in a PARP1 and TIMELESS-dependent manner. The study manages to integrate this specific function with a role of chromatin dynamics in transcription repression at the site of damage and recombination enhancement as a DSB repair pathway. Authors make interesting proposals in their Discussion that would certainly need to be tested with further experiments. For example, it in the future would be nice to know how the FBXL10-RNF68-RNF2 complex affect DSB resection, and other aspects. However, having said this, this is manuscript seems suitable for *eLife*.

Reviewer #2:

This manuscript from Pagano and colleagues reports a role for a non-canonical PRC1 complex, consisting of Fbxl10, Rnf2, and Rnf68 in the DNA double strand break (DSB) response. They show by STORM imaging that the complex localizes to DSBs and that is facilitates ubiquitination of H2A on K119, although it is unclear if this ubiquitination is direct, or mediated by the recruitment of a second E3 ubiquitin ligase. Further they show that the complex represses transcription at sites of DNA damage, and that this facilitates incorporation of H2A.Z and ultimately in successful homologous recombination repair. Overall, the manuscript is well written and the data is of the highest quality.

1) STORM imaging suggests that the entire complex is localized to DNA damage repair sites in response to NCS. In previous studies these authors have shown the number of overlaps per nuclei with and without damage (PNAS 2015). They should do the same thing here to further convince readers that the overlap is real, particularly in light of the fact that this is a non-conventional way of showing these correlations that many readers will be unfamiliar with.

2) In Figure 1—figure supplement 2 the authors show GFP tagged components forming foci in response to damage ("damage spots"), although all of these proteins are forming foci without damage and it is unclear if these proteins would blink on and off with no treatment. How do the authors know these are sites of DNA damage? This should be addressed.

3) The authors show that ATM inhibition restores active transcription at sites of DNA damage. This result is consistent with data from Shanbhag et al. (2010) showing that ATM is required to repress transcription at sites of DNA damage. However, they also show that ATM inhibition has no effect on the localization of Fbxl10, Rnf2, and Rnf68. Thus, it is unclear how ATM inhibition is repressing transcription if the Fbxl10 complex is still localized. The authors should provide some explanation for how this might occur.

4) In Figures 5 and 6 the authors make the argument that the role of this complex is dependent on deposition of H2A.Z at sites of damage. They should again quantify the number of overlaps per cell.

5) It is unclear what the timing of H2A.Z deposition is in relation to that of Fbxl10, Rnf2, and Rnf68. The latter cycles through break sites very early (within minutes), well before the deposition of Rad51 onto single stranded DNA.

6) It is also unclear how H2A.Z is being deposited, just that it depends on Fbxl10, Rnf2, and Rnf68. Therefore, this section of the manuscript feels mis-placed and premature. While it appears that H2a.Z is implicated in the DNA damage response and HRR, it is unclear how or why, and how this is connected to Fbxl10, Rnf2, and Rnf68, just that there exists a dependency.

Reviewer #3:

The manuscript entitled "PARP1-dependent recruitment of the FBXL10-RNF68-RNF2 ubiquitin ligase to sites of DNA damage controls H2A.Z loading" by Pagano et al., describes the role of the non-canonical PRC1 complex FBXL10-RNF68-RNF2 in DNA damage response and repair via transcription repression and H2AZ exchange at sites of DNA damage. The authors report that this ncPRC1.1 complex, which is recruited in a PARP dependent manner, mediates H2AK119Ub and promotes HR repair. The experiments performed using depletion of the individual subunits of the complex suggests the involvement of these proteins in DNA damage responses. However, functional importance of this complex, for instance, a mechanistic insight of how this complex promotes H2AZ incorporation is lacking. In addition, given the published literature on the canonical PRC1 complex being involved in transcriptional repression at DNA damage sites (ex. Sanchez et al., 2016), it is critical to determine whether the reported non-canonical PRC1 complex works in the same pathway or delineate the interplay between canonical PRC1 complex activity at damage sites and this non-canonical PRC1 complex. As is, this story is rather descriptive and lacks mechanistic insights.

1) In Figure 1, it is unclear why the untreated U2OS cells (-NCS) also show high levels of XRCC5. FBXL10, RNF68, RNF2 recruitment are shown to be PARP1 and TIMELESS dependent manner but it is unclear if this complex works collectively. Given that these proteins are involved in other complexes known to be recruited to chromatin, are FBXL10, RNF68 and RNF2 DSB recruitment interdependent or independent of each other? How do these proteins affect the canonical PRC complexes? It will also be informative to check for the recruitment of FBXL10, RNF68, RNF2 in the RNF68 RING domain and Y252D mutant.

- The authors mention that RNF2 remains at DNA lesions longer likely because of its presence in canonical and ncPRC complex. RNF2 is known to interact with BMI1 that has been shown to recruit and be retained at DSBs for several hours. It would be interesting to check if depletion of BMI1 affects RNF2 recruitment kinetics in this context.

- It is reported that only PARP1 and TIMELESS completely abrogate FBXL10, RNF68 and RNF2 DSB recruitment but from Figure 1—figure supplement.2D it is clear that ATM kinase inhibition shows strong DSB recruitment and retention (RNF68 in particular). Is there an explanation for this result?

2) In Figure 2, it is shown that depletion of FBXL10, RNF68 and RNF2 reduces H2A K119Ub1 suggesting FBXL10-RNF68-RNF2 as the E3 ligase complex. However it has been previously reported that H2AK119Ub1 at DNA damage site is mediated by BMI1-RNF2 that makes it unclear about the specificity of this ncPRC1 at the level of the core PRC complexes. The authors show that FBXL10-RNF68-RNF2 is recruited early in the DNA damage response. On the other hand BMI1 is recruited later in the DDR. It will be interesting to check if FBXL10-RNF68-RNF2 mediates the first wave of H2AK119ub1 while BMI1-RNF2 mediates subsequent H2AK119Ub1. Moreover, it will be also useful to check for H2AK119Ub1 in RNF68 RING domain and Y252D mutants.

3) It would be appropriate to show a comparison of H2A/H2AX119Ub -/+ DNA damage in Control vs. FBXL10, RNF68 and RNF2 deficient cells by western blot.

4) There is a possibility that depletion of FBXL10-RNF68-RNF2 affects other histone ubiquitylation marks in DDR given that in Figure 4A-B, reduction in polyubiquitin signal (FK2) was observed. It is unclear if other such Ub marks on histones were analyzed in this study. DNA damage induced ubiquitin foci (detected by FK2) are formed mostly by RNF8 and RNF168. Do the authors suggest that FBXL10-RNF68-RNF2 works upstream or negatively affects RNF8 and RNF168 or its substrates and if so how? It will be interesting to check if FBXL10-RNF68-RNF2 affects RNF8 recruitment or vice versa.

5) It is important to perform cell survival assays in the context of DDR as previous reports state that RNF2 reduces cell viability and promotes cell cycle arrest.

6) It is unclear whether the demethylation activity of Fbxl10 is involved in its role in the DDR demonstrated in this article i.e., H2AK119ub1, transcriptional repression, 53BP1 and RAD51 recruitment? Demethylation of H3K4me3 has been shown to be involved in these pathways so this is worth considering.

7) It should be explained why there are phenotypical differences observed when depleting FBXL10, RNF68 and RNF2 if these form a functional complex. For example, why FBXL10 shows NHEJ defects while the other members do not? Moreover, another piece of work has previously reported that loss of H2AZ decreases Ku-loading and NHEJ but this doesn't fit well with the experimental results shown here (Price and colleagues, Mol Cell, 2012). There is no mechanistic insights for how this complex promotes H2AZ accumulation at damage sites. From the IF images, clearly most H2AZ incorporation is not affected by these proteins (see comment 9). Most studies need to be performed to understand these preliminary data.

8) Complementation studies are missing, which could significantly strengthen the data.

9) It would be appropriate to include the focal accumulation images of the DDR repair factors along with the quantification shown in Figure 4. Instead of representing the% cells with foci, the number of foci per cell would be more informative. The microscopy data presented for high resolution has some major issues. As colocalization is used, it is unclear if these proteins really accumulate at IR induced damage sites or if this is just due to the increased signal obtained by KU or phosphH2AX. If you look at just the signal of the non-canonical PRC1 proteins, it is unclear if the signals increase and/or the number of foci change. These parameters must be analyzed to ensure that this technique is not an artifact of the system and how these data are being presented. As is, these data are not convincing.

---

## [Author Response]

[Editors’ note: the author responses to the first round of peer review follow.]

This manuscript describes a series of complex and high-quality experiments using a varied combination of cell biology and molecular biology approaches to conclude that the core ubiquitin ligase subcomplex of the ncPRC1.1 Polycomb repressive complex formed by FBXL10, RNF2 and RNF68 is recruited to DNA double strand breaks in a PARP1 and TIMELESS dependent manner. Recruitment is accompanied by mono-ubiquitylation of H2A K119, consistent with the activity of the FBXL10-RNF2-RNF68 complex, and a reduction of H2A density that is accompanied by an increase of H2AZ. As a consequence of these changes, transcription is reduced at the site of damage. Finally, the authors show that FBXL10-RNF2-RNF68 is required for efficient homologous recombination repair of the DSB. The study thus provides a general picture of the control of HR DSB repair by FBXL10-RNF68-RNF2 complex in a PARP1 and TIMELESS-dependent manner, but a mechanistic insight of how this complex promotes H2AZ incorporation is lacking. Despite the potential and interest of the work for eLife, the H2A.Z part, critical for the novelty of the manuscript, should be developed further. It would be essential to figure out the connection between ncPRC and cPRC and H2AZ deposition. Additionally, the STORM data does not seem sufficiently valid as it is and needs additional work to show the claims. More specific comments can be found in the individual reviews attached below.Reviewer #2:[…] 1) STORM imaging suggests that the entire complex is localized to DNA damage repair sites in response to NCS. In previous studies these authors have shown the number of overlaps per nuclei with and without damage (PNAS 2015). They should do the same thing here to further convince readers that the overlap is real, particularly in light of the fact that this is a non-conventional way of showing these correlations that many readers will be unfamiliar with.

Figure 1A shows representative overlaps between FBXL10, RNF68, or RNF2 and XRCC5, with or without NCS treatment. However, as now mentioned in the manuscript, in dense images, it is possible that two species randomly overlap with each other. Therefore, we analyzed all images using a cross-correlation method that probes the probability distributions across all possible pair-wise distances between two species taking in account, at the same time, the amount of each species (PMIDs: 25179006, 23717596, 22384026 and 27545293). Briefly, if two species are randomly distributed from each other, the probability should be the same at all distances (i.e., the probability should be flat if visualized as a function of the pairwise distances). In contrast, if the two species have intrinsic spatial correlation at a certain distance, the correlation function will display a significant probability distribution at such certain distance (although such correlations are not easily seen because of their stochastic and dense distributions). We have now explained in more detail in the text the benefit of using super-resolution microscopy and the way data was analyzed. Moreover, we have now also used confocal microscopy as an orthogonal method to show the enrichment of the endogenous FBXL10-RNF68-RNF2 ubiquitin ligase complex (FRRUC) at sites of DNA damage (new Figure 1—figure supplement 1B).

2) In Figure 1—figure supplement 2 the authors show GFP tagged components forming foci in response to damage ("damage spots"), although all of these proteins are forming foci without damage and it is unclear if these proteins would blink on and off with no treatment. How do the authors know these are sites of DNA damage? This should be addressed.

The damage spots are only those indicated by the white dotted lines (these are the spots we hit with the laser prior to live imaging). The other large spots in cells expressing FBXL10 and RNF68 are nucleoli. FBXL10 is known to predominantly localize to the nucleolus (PMIDs: 17994099 and 19852816). Our data show that RNF68 displays a similar localization, while RNF2, which is present in several protein complexes, has a more homogenous distribution. Many nucleolar proteins involved in the DNA damage response are sequestered in the nucleolus (PMIDs: 29429989, 23879289, etc.).

3) The authors show that ATM inhibition restores active transcription at sites of DNA damage. This result is consistent with data from Shanbhag et al. (2010) showing that ATM is required to repress transcription at sites of DNA damage. However, they also show that ATM inhibition has no effect on the localization of Fbxl10, Rnf2, and Rnf68. Thus, it is unclear how ATM inhibition is repressing transcription if the Fbxl10 complex is still localized. The authors should provide some explanation for how this might occur.

The FRRUC is recruited to DNA lesion independently of ATM (Figure 1—figure supplement 2D). Thus, they seem to function in parallel pathways and, in fact, ATM seems to have an inhibitory role on the recruitment of the FRRUC. ATM might limit the spread of the FRRUC and the subsequent H2A mono-ubiquitylation to maintain the signal at DSB proximities. We now discuss this point (Discussion, fifth paragraph).

4) In Figures 5 and 6 the authors make the argument that the role of this complex is dependent on deposition of H2A.Z at sites of damage. They should again quantify the number of overlaps per cell.

Please see our reply to point #1.

5) It is unclear what the timing of H2A.Z deposition is in relation to that of Fbxl10, Rnf2, and Rnf68. The latter cycles through break sites very early (within minutes), well before the deposition of Rad51 onto single stranded DNA.

H2A.Z recruitment is detectable within two minutes after DSB induction and is removed within 10 minutes (PMIDs: 26142279 and 26034280). Unfortunately, we were unable to show H2A.Z recruitment to laser damage, possibly because of the very dense H2A.Z staining. However, we were able to show H2A.Z recruitment by both ChIP and super-resolution microscopy (Figure 6A-C). Interestingly, a recent paper measured the kinetics of recruitment of 70 DNA repair proteins to laser-induced DNA damage sites in HeLa cells (PMID: 29547717). Compared to the recruitment of these 70 proteins, FBXL10, RNF68, and RNF2 display a very fast recruitment while RAD51’s peak of recruitment is about 18 minutes. The rapid recruitment of the FRRUC suggests that this represent one of the most upstream events in response to genotoxic stress, affecting many downstream events of which some directly and others indirectly.

6) It is also unclear how H2A.Z is being deposited, just that it depends on Fbxl10, Rnf2, and Rnf68. Therefore, this section of the manuscript feels mis-placed and premature. While it appears that H2a.Z is implicated in the DNA damage response and HRR, it is unclear how or why, and how this is connected to Fbxl10, Rnf2, and Rnf68, just that there exists a dependency.

We have now gained mechanistic insights into how the FRRUC promotes the recruitment of H2A.Z (new Figure 6F and new Figure 6—figure supplement 1G-I). Specifically, we now provide evidence that, whereas H2A.Z is not required for H2A mono-ubiquitylation, the latter is directly necessary for the enrichment of H2A.Z at DNA damage sites. To reach this conclusion, we utilized a strategy used by a landmark Cell paper on DNA damage-induced transcription repression (PMID: 20550933). As in this study, either wild-type H2A or H2A(K119/120R), an H2A mutant in which Lys119 and Lys120 were mutated to arginines, were overexpressed in the reporter cells expressing inducible FokI. We observed a reduction in H2A.Z loading at DSBs in cells overexpressing H2A(K119/120R) compared to cells overexpressing wild-type H2A (new Figure 6F). This is in agreement with the fact that only wild-type RNF68 (able to sustain H2A mono-ubiquitylation), but not two inactive RNF68 mutants, rescues the defect of H2A.Z loading (Figure 6E), and indicates that the FRRUC-dependent mono-ubiquitylation of H2A is required for proper H2A.Z loading. In contrast, the chaperones that contribute to H2A.Z loading (i.e., TIP48 and TIP49) are not controlled by the FRRUC (Figure 6—figure supplement 1G-H). H2A.Z loading has been previously shown to be involved in DNA repair mechanisms. We further show that H2A.Z loading depends on the FRRUC and contributes to transcriptional repression and the DDR (Figures 6 and 7). How H2A.Z recruitment is involved in transcriptional repression is currently unclear. But this is true also for modifications that are commonly accepted to be necessary for transcriptional repression. For example, H2A mono-ubiquitylation is thought to repress transcription; however, how mono-ubiquitylation of histones mechanistically controls downstream effects (such as nucleosome rearrangements and chromatin remodeling) remains a big, black box in both the field of DNA damage and the field of transcription.

Reviewer #3:The manuscript entitled "PARP1-dependent recruitment of the FBXL10-RNF68-RNF2 ubiquitin ligase to sites of DNA damage controls H2A.Z loading" by Pagano et al., describes the role of the non-canonical PRC1 complex FBXL10-RNF68-RNF2 in DNA damage response and repair via transcription repression and H2AZ exchange at sites of DNA damage. The authors report that this ncPRC1.1 complex, which is recruited in a PARP dependent manner, mediates H2AK119Ub and promotes HR repair. The experiments performed using depletion of the individual subunits of the complex suggests the involvement of these proteins in DNA damage responses. However, functional importance of this complex, for instance, a mechanistic insight of how this complex promotes H2AZ incorporation is lacking. In addition, given the published literature on the canonical PRC1 complex being involved in transcriptional repression at DNA damage sites (ex. Sanchez et al., 2016), it is critical to determine whether the reported non-canonical PRC1 complex works in the same pathway or delineate the interplay between canonical PRC1 complex activity at damage sites and this non-canonical PRC1 complex. As is, this story is rather descriptive and lacks mechanistic insights.1) In Figure 1, it is unclear why the untreated U2OS cells (-NCS) also show high levels of XRCC5.

XRCC5 protein levels do not change after DNA damage, only its localization. XRCC5 is a highly abundant protein that is rapidly recruited to DSBs since, although it has intrinsic affinity for DNA, it has even higher affinity for DNA ends. Because it associates with chromatin in both undamaged and damaged cells (although at different locations under these two conditions), XRCC5 is an ideal marker to evaluate cross-correlation changes with other DNA-binding proteins in response to DNA damage (PMIDs: 23897892, 25941401, 29215009, 11493912). We have now clarified the main text.

FBXL10, RNF68, RNF2 recruitment are shown to be PARP1 and TIMELESS dependent manner but it is unclear if this complex works collectively. Given that these proteins are involved in other complexes known to be recruited to chromatin, are FBXL10, RNF68 and RNF2 DSB recruitment interdependent or independent of each other?

The hierarchical nature of the FBXL10-RNF68-RNF2 ubiquitin ligase complex (FRRUC) has been well established using both structural biology and biochemical approaches: its chromatin recruitment depends on FBXL10, which recruits RNF68, which, in turn, recruits RNF2 (PMIDs: 27568929, 23523425, 24856970, 23256043, 23502314, 23395003). Accordingly, silencing of FBXL10 impairs the recruitment of RNF68 and to some extent RNF2 (see Author response image 1), the latter also being recruited via the cPRC1. Moreover, we confirm that FBXL10, RNF68, and RNF2 form a trimeric complex using sequential immunoprecipitations (new Figure 1—figure supplement 1A).

**Author response image 1. respfig1:** U2OS cells were transfected with the indicated siRNAs. After 72 h, cells were treated with NCS for 20 minutes. After harvesting, cells were fractionated into soluble and chromatin fractions, and lysates were immunoblotted as indicated. For chromatin fractionation, cells were pre-extracted with CSK buffer (D’Angiolella et al., 2010) for 10 minutes on ice and centrifuged for 3 min at 3000 g. The insoluble pellets were digested with TurboNuclease in the presence of 250 mM NaCl to generate the chromatin fraction. Asterisks denote unspecific bands.

How do these proteins affect the canonical PRC complexes? It will also be informative to check for the recruitment of FBXL10, RNF68, RNF2 in the RNF68 RING domain and Y252D mutant.- The authors mention that RNF2 remains at DNA lesions longer likely because of its presence in canonical and ncPRC complex. RNF2 is known to interact with BMI1 that has been shown to recruit and be retained at DSBs for several hours. It would be interesting to check if depletion of BMI1 affects RNF2 recruitment kinetics in this context.

We have now included data on the recruitment of RNF51 (AKA BMI1 or PCGF4) and RNF110 (AKA MEL18 or PCGF2), which are components of cPRC1 complexes, and found a dependence on PARP1 and TIMELESS (new Figure 2B-C and new Figure 2—figure supplement 1B). Notably, their recruitment is inhibited when RNF68 is depleted. Wild-type RNF68, but not RNF68 inactive mutants, rescue this phenotype (new Figure 2B-C). In contrast, recruitment of FBXL10 and RNF68 is not affected by the silencing of either RNF51 or RNF110 (new Figure 2A and new Figure 2—figure supplement 1A). Thus, it appears that the FRRUC is necessary for the proper recruitment of the cPRC1 complexes to DNA lesions.

- It is reported that only PARP1 and TIMELESS completely abrogate FBXL10, RNF68 and RNF2 DSB recruitment but from Figure 1—figure supplement 2D it is clear that ATM kinase inhibition shows strong DSB recruitment and retention (RNF68 in particular). Is there an explanation for this result?

ATM might limit the spread of the FRRUC and the subsequent H2A mono-ubiquitylation to maintain the signal at DSB proximities. We now discuss this point (Discussion, fifth paragraph).

2) In Figure 2, it is shown that depletion of FBXL10, RNF68 and RNF2 reduces H2A K119Ub1 suggesting FBXL10-RNF68-RNF2 as the E3 ligase complex. However it has been previously reported that H2AK119Ub1 at DNA damage site is mediated by BMI1-RNF2 that makes it unclear about the specificity of this ncPRC1 at the level of the core PRC complexes. The authors show that FBXL10-RNF68-RNF2 is recruited early in the DNA damage response. On the other hand BMI1 is recruited later in the DDR. It will be interesting to check if FBXL10-RNF68-RNF2 mediates the first wave of H2AK119ub1 while BMI1-RNF2 mediates subsequent H2AK119Ub1. Moreover, it will be also useful to check for H2AK119Ub1 in RNF68 RING domain and Y252D mutants.

It appears that the FRRUC is necessary for the proper recruitment of the cPRC1 complexes to DNA lesions (new Figure 2 and new Figure 2—figure supplement 1-2). See point #1 for details.

3) It would be appropriate to show a comparison of H2A/H2AX119Ub -/+ DNA damage in Control vs. FBXL10, RNF68 and RNF2 deficient cells by western blot.

H2AK119Ub1 is a widespread epigenetic marker modulating transcription in unperturbed cells. Upon DNA damage, this signal is locally enriched at DNA damage sites. We were unable to reliably show this local enrichment by immunoblotting, thus, we have not included these results in the manuscript. However, we have shown the local enrichment of H2AK119Ub1 with two independent methods: (i) using local laser micro-irradiation and immunofluorescence staining (Figure 3A-B) and (ii) by ChIP analysis at locally introduced DSBs using a nuclease (Figure 3C).

4) There is a possibility that depletion of FBXL10-RNF68-RNF2 affects other histone ubiquitylation marks in DDR given that in Figure 4A-B, reduction in polyubiquitin signal (FK2) was observed. It is unclear if other such Ub marks on histones were analyzed in this study. DNA damage induced ubiquitin foci (detected by FK2) are formed mostly by RNF8 and RNF168. Do the authors suggest that FBXL10-RNF68-RNF2 works upstream or negatively affects RNF8 and RNF168 or its substrates and if so how? It will be interesting to check if FBXL10-RNF68-RNF2 affects RNF8 recruitment or vice versa.

As the reviewer suggested, we have measured the recruitment kinetics of GFP-RNF8 (Author response image 2). By 180 seconds, RNF8 had not reached its maximal level of recruitment. Silencing FBXL10, RNF68, or RNF2 negatively affected the recruitment of RNF8. The time of RNF8 recruitment is much slower than what we measure for the FRRUC, which is ~60 seconds. We believe that H2AK119Ub1 is one of the earliest markers of DSB and, therefore, it affects many late events (directly and indirectly), including the recruitment of RNF8.

**Author response image 2. respfig2:** U-2OS cells stably expressing GFP-RNF8 (a kind gift from Dr D Durocher) were transfected with siRNAs targeting FBXL10, RNF68, RNF2 or H2A.Z, or a non-targeting control (CTRL). Cells were pre-sensitized with BrdU (10 μM) for 36 hours and subjected to 405 nm laser induced damage. DNA damage recruitment dynamics were captured by live cell imaging. Relative fluorescence values and images were acquired every 5 seconds for 3 minutes. For each condition, ≥25 cells were evaluated from 2 or 3 independent experiments. Mean relative fluorescence values and standard errors were plotted against time.

5) It is important to perform cell survival assays in the context of DDR as previous reports state that RNF2 reduces cell viability and promotes cell cycle arrest.

We now show that the silencing of FBXL10, RNF68, or RNF2 sensitizes cells to DNA damage (new Figure 5—figure supplement 2F).

6) It is unclear whether the demethylation activity of Fbxl10 is involved in its role in the DDR demonstrated in this article i.e., H2AK119ub1, transcriptional repression, 53BP1 and RAD51 recruitment? Demethylation of H3K4me3 has been shown to be involved in these pathways so this is worth considering.

FBXL10 display multiple splice isoforms, but the FRRUC within the ncPRC1.1 contains only a short isoform lacking the N-terminal JmjC histone demethylase domain (PMIDs: 25533466, 26687479, 17296600, 29502955, etc.).

7) It should be explained why there are phenotypical differences observed when depleting FBXL10, RNF68 and RNF2 if these form a functional complex. For example, why FBXL10 shows NHEJ defects while the other members do not? Moreover, another piece of work has previously reported that loss of H2AZ decreases Ku-loading and NHEJ but this doesn't fit well with the experimental results shown here (Price and colleagues, Mol Cell, 2012).

Depletion of FBXL10 results in a reduction in NHEJ, in agreement with the reduction in 53BP1 foci (Figure 5—figure supplement 1A-B and Figure 5—figure supplement 2B-C), suggesting that FBXL10 may play a role in NHEJ, independently of RNF68 and RNF2. This is in agreement with the finding that FBXL10 (which has several isoforms and whose short isoform is the only one participating to the FRRUC) can participate in several protein complexes (PMID: 17296600). We have now better explained this in the text. Price and colleagues showed (PMID: 23122415) that H2A.Z incorporation is necessary for both HRR and NHEJ, as both pathways were significantly inhibited upon depletion of H2A.Z. Interestingly, in our hands we barely see a reduction in NHEJ repair efficiency upon depletion of H2A.Z (Figure 5—figure supplement 2E).

There is no mechanistic insights for how this complex promotes H2AZ accumulation at damage sites. From the IF images, clearly most H2AZ incorporation is not affected by these proteins (see comment 9). Most studies need to be performed to understand these preliminary data.

Figure 6A-B and D-E shows that H2A.Z incorporation is significantly reduced by the silencing of FBXL10, RNF68 or RNF2. We have now gained mechanistic insights into how the FRRUC promotes the recruitment of H2A.Z (new Figure 6F and new Figure 6—figure supplement 1G-I). Specifically, we now provide evidence that, whereas H2A.Z is not required for H2A mono-ubiquitylation, the latter is directly necessary for the enrichment of H2A.Z at DNA damage sites. To reach this conclusion, we utilized a strategy used by a landmark *Cell* paper on DNA damage-induced transcription repression (PMID: 20550933). As in this study, either wild-type H2A or H2A(K119/120R), an H2A mutant in which Lys119 and Lys120 were mutated to arginines, were overexpressed in the reporter cells expressing inducible FokI. We observed a reduction in H2A.Z loading at DSBs in cells overexpressing H2A(K119/120R) compared to cells overexpressing wild-type H2A (new Figure 6F). This is in agreement with the fact that only wild-type RNF68 (able to sustain H2A mono-ubiquitylation), but not two inactive RNF68 mutants, rescues the defect of H2A.Z loading (Figure 6E), and indicates that the FRRUC-dependent mono-ubiquitylation of H2A is required for proper H2A.Z loading. In contrast, the chaperones that contribute to H2A.Z loading (*i.e.,* TIP48 and TIP49) are not controlled by the FRRUC (new Figure 6—figure supplement 1G-H).

8) Complementation studies are missing, which could significantly strengthen the data.

In order to control for off-target effects, we sought to rescue observed phenotypes by complementation of siRNA-resistant constructs. We used both wild-type RNF68 and inactive RNF68 mutants and complemented the phenotypes observed upon RNF68 silencing (new Figure 2B-C, new Figure 3B, Figure 5C and F, Figure 6E). We focused on rescuing the effect of RNF68 depletion since RNF68 is the subunit that defines the ncPRC1.1 complex (PMID: 23523425). Notably, we show that the *entire* FRRUC is needed for proper H2A.Z loading, DDR signaling and H2A mono-ubiquitination since the silencing of RNF68 could not be rescued with either a mutant unable to bind FBXL10 (Y252D) or a mutant that is not able to bind RNF2 (ΔRING) (Figure 5C and F, Figure 6E, new Figure 3B).

9) It would be appropriate to include the focal accumulation images of the DDR repair factors along with the quantification shown in Figure 4.

Representative images for the focal accumulation of DDR repair factors were in the supplement (now in Figure 5—figure supplement 1A-B).

Instead of representing the% cells with foci, the number of foci per cell would be more informative.

The percentage of cells containing DNA-damage induced foci were evaluated as described in several landmark papers in the DNA damage field by normalizing to the basal level of genotoxic stress present in cultured cells (e.g., PMID: 20362325, 23333306, 26649820). Therefore, we believe that thresholding foci formation to a certain amount of foci of DDR foci per cell is a sensitive way to measure impairment in signaling pathway assembly.

The microscopy data presented for high resolution has some major issues. As colocalization is used, it is unclear if these proteins really accumulate at IR induced damage sites or if this is just due to the increased signal obtained by KU or phosphH2AX. If you look at just the signal of the non-canonical PRC1 proteins, it is unclear if the signals increase and/or the number of foci change. These parameters must be analyzed to ensure that this technique is not an artifact of the system and how these data are being presented. As is, these data are not convincing.

We analyzed all super-resolution images using a cross-correlation method that probes the probability distributions across all possible pair-wise distances between two species taking in account, at the same time, the amount of each species (PMIDs: 25179006, 23717596, 22384026 and 27545293). Therefore, the given cross-correlation was already normalized by the amount of the species, to make sure the increase in cross-correlation signal is not due to the increase in the amount of either specie but their real co-localization probability density. In any case, in response to DNA damage, XRCC5 or FBXL10, RNF68 and RNF2 localize to DSB, but their levels do not change. Moreover, when we downregulate FBXL10, RNF68, or RNF2, we observe no changes in XRCC5 and phospho-H2A.X levels, compared to control cells (new Figure 3—figure supplement 1B, Figure 5D and new Figure 5—figure supplement 2A). We have now clarified the text.